# HTS Accelerator Magnet and Conductor Development in Europe

**Lucio Rossi** [1,2,3,*] **and Carmine Senatore** [4]

1   Physics Department, Università di Milano, Via Fratelli Cervi, 201, 20054 Segrate (Milano), Italy
2   INFN-LASA-Milano, Via Fratelli Cervi, 201, 20054 Segrate (Milano), Italy
3   CERN, Accelerator & Technology Sector, 1211 Geneva, Switzerland
4   Department of Quantum Matter Physics (DQMP), Université de Genève, 1211 Geneva, Switzerland; carmine.senatore@unige.ch
*   Correspondence: lucio.rossi@cern.ch

**Abstract:** In view of the preparation for a post-LHC collider, in 2010 the high-energy physics (HEP) community started to discuss various options, including the use of HTS for very high-field dipoles. Therefore, a small program was begun in Europe that aimed at exploring the possibility of using HTS for accelerator-quality magnets. Based on various EU-funded programs, though at modest levels, it has enabled the European community of accelerator magnet research to start getting experience in HTS and address a few issues. The program was based on the use of $REBa_2Cu_3O_{7-x}$ (REBCO) tapes to form 10 kA Roebel cables to wind small dipoles of 30–40 mm aperture in the 5 T range. The dipoles are designed to be later inserted in a background dipole field (in $Nb_3Sn$), to reach eventually a field level in the 16–20 T range, beyond the reach of Low Temperature Superconductors (LTS). The program is currently underway: more than 1 km of high-performance tape ($J_e > 500 \ A/mm^2$ at 20 T, 4.2 K) has been manufactured and characterized, various 30 m long Roebel cables have been assembled and validated up to 13 kA, a few dipoles have been wound and tested, reaching 4.5 T in stand-alone (while a dipole made from flat race track coils exceeded 5 T using stacked tape cable), and tests in background field are being organized.

**Keywords:** high current density HTS; Roebel cables; accelerator magnets; collider magnets; superconducting HTS magnets

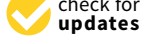



## 1. Introduction

Accelerators have been among the most powerful instruments for scientific discoveries. The identification of the Higgs boson [1,2] at CERN on 4 July 2012 was the first big discovery of the LHC, the largest instrument for high-energy physics, which was put in operation in 2008 and then operated in collider mode from March 2010. The discovery of the Higgs boson and its characterization has opened new perspectives for physics beyond the standard model. While the LHC is striving to reach its design beam energy by pushing the Nb-Ti superconducting magnets [3] to their design value of 8.3 T in the next LHC Run 3 of 2021–2024, the HEP (high-energy physics) accelerator community is actively pursuing the next step: the high-luminosity LHC (HL-LHC or HiLumi) machine [4,5], for which a new generation of superconducting accelerator magnets is now under construction [6,7]. Meanwhile, the next step is preparing to go even further with magnet technology.

## 2. Accelerator and Magnets

The discovery potential of a hadron collider depends critically on the strength of the magnetic field in various ways. In a circular accelerator or collider, the energy of the particle beam depends directly on the size of the accelerator and on the main dipole field according to the simple relation (valid for relativistic particles) $E \cong 0.3 \ B \ R$, where E is the beam energy in TeV (teraelectronvolt), B is the field of the dipoles in tesla, and R is the

effective radius of the accelerator in km. In the LHC, the 8.3 T dipole field with a 2.8 km effective radius (physical radius is 4.2 km, however the bending field covers only 2/3 of the 26.7 km tunnel length) yielding 7 TeV beam energy, allowing 14 TeV in the center of mass during collisions.

There is another important figure of merit to qualify a collider as a discovery instrument: the luminosity—i.e., the collision rate per unit of reaction cross-section. It is not enough to produce only one Higgs boson. Plenty of them must be generated in order to allow their identification and to measure subtle properties. In a field like HEP, where the interesting phenomena are more and more rare, surrounded by billions of "useless" interactions (noise), luminosity—i.e., the quantity of useful particles produced during collision that can be detected—is becoming of paramount importance. That is why CERN is investing more than 1 B\$ (material budget only), which is 25–30% of the LHC cost, just to improve the luminosity with the above-mentioned HL-LHC project. It turns out that luminosity depends on the peak field of special magnets in the low-beta insertion. In particular, the so-called Inner Triplet (IT) quadrupoles, controlling the size of the beam (beta) at the collision, are the most critical magnets in terms of peak field, since they feature both high field gradients and large apertures. For the HL-LHC, peak fields in the range 11–12 T are then required for large-aperture IT quadrupoles for the previously mentioned reasons. In addition, also a few dipoles need to be designed at 11 T in order to accommodate beam cleaning collimators in the cold region. These collimators are necessary for increasing the beam intensity (another important factor to improve luminosity). Therefore, the high-luminosity LHC, with its magnets that are 30–40% more powerful than LHC ones, is a novel step in improving the LHC as a discovery instrument. These magnets constitute a breakthrough in accelerator technology; indeed, this can be seen in Figure 1, which reports the progress in the magnetic field for past and future accelerators.

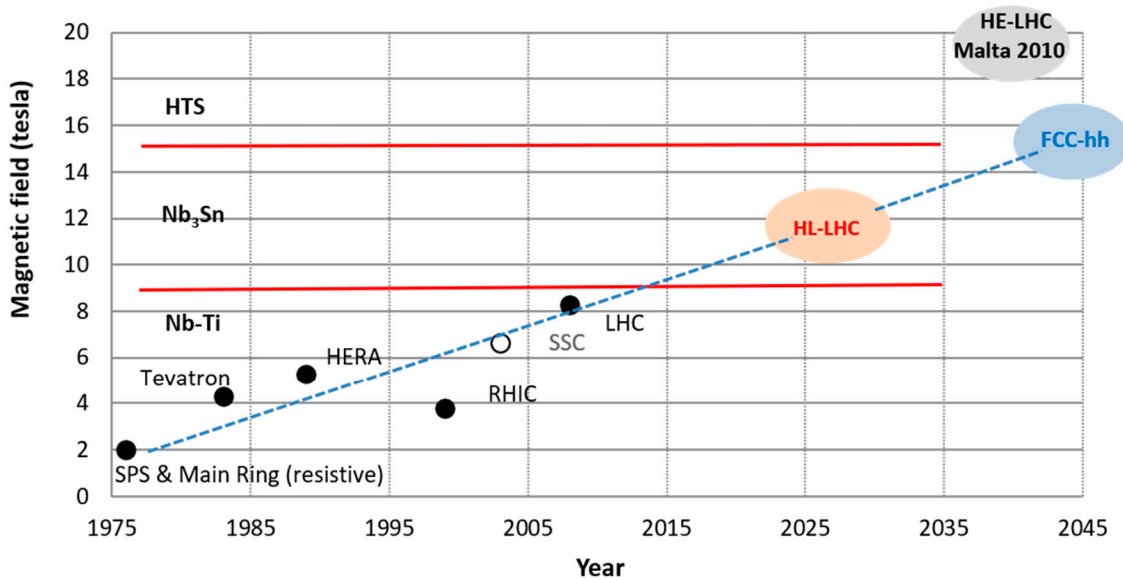

**Figure 1.** Progress of the operating field in circular hadron colliders (ill-fated SSC, i.e., the Superconducting Super Collider, is reported for comparison). LHC dipoles are actually working at 7.7 T and are supposed to operate at 8.3 T design field in 2022–2023. HL-LHC has already shown its field level in a few prototypes. HE-LHC (Malta workshop 2010) is a pure paper design. FCC-hh started dedicated R&D in 2014.

As a further step beyond LHC and its high-luminosity upgrade, in 2010 CERN investigated an energy upgrade of the LHC, called high-energy LHC (HE-LHC) [8], based on filling the present 27 km-long LHC tunnel with 20 T dipole magnets [9]; see Figure 1. Based on the use of HTS (High Temperature Superconductors) for the inner coil layer and

LTS for the outer coil layers, this project, also in view of the lengthy R&D for the 12 T field level of HL-LHC, soon appeared to be far too ambitious with respect to the technology and not ambitious enough in its energy reach (33 TeV center-of mass energy). In 2013, based on the outcome of the new update of the European Strategy of Particle Physics, as a possible next step post-LHC and post-HiLumi, CERN proposed an 80–100 km circular collider called FCC (Future Circular Collider) that would reach a 100 TeV center-of-mass energy. This would require the dipole field to reach 20 T (for 80 km) or 16 T (for 100 km) in operation [10]. Soon, the baseline of the dipole field value was fixed to be 16 T, to stay possibly within the reach of $Nb_3Sn$; see Figure 1. HTS was clearly perceived as a technology not yet mature enough for accelerators, and for good reason. It appears evident on the plot of Figure 1 that going to 15–16 T would have been a huge jump. However, the need of HTS for use in a next higher-energy collider either for main dipoles in the far future or in special regions of FCC (of high-radiation environment and/or high thermal load) was clear. In addition, it was also evident that only dedicated R&D could determine if HTS could meet the very demanding specifications for collider magnets. For this reason, a continuous, though modest, effort started in Europe in 2012–2013 devoted to accelerator magnets using HTS.

### 3. The FP7-EuCARD HTS Racetrack Magnet

Well before the above-mentioned program for FCC, in 2007 an integrated research activity for accelerator R&D was launched, applying for EU funding in the program FP7 with the name FP7-EuCARD (European Coordination of Accelerator R&D). The EuCARD magnet work package was mainly devoted to developing conductor and magnet technology for the HiLumi LHC magnets, featuring the design and construction of the Fresca2 large bore dipole, with 100 mm of free coil aperture and a 13 T nominal field at 1.9 K, wound in $Nb_3Sn$. Fresca2 did reach 14.6 T [11], which is at present the record field for a dipole magnet. See Figure 2 for a schematic view of Fresca2 and its training curve.

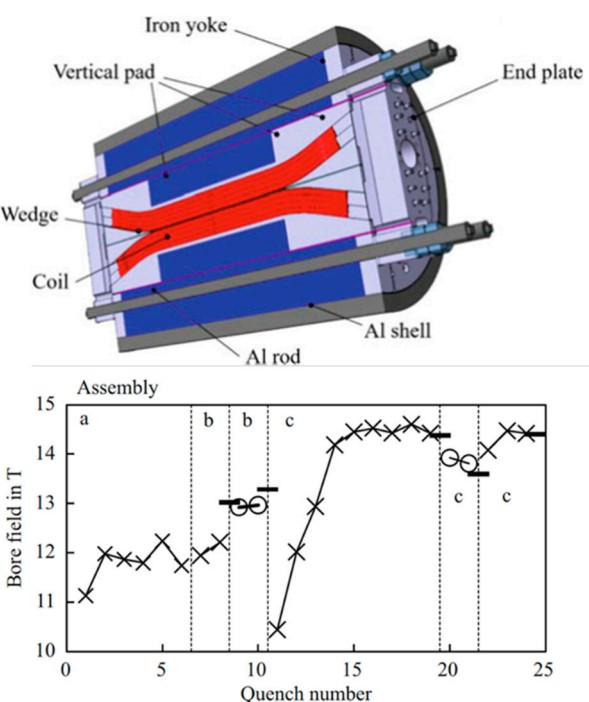

**Figure 2.** 3D sketch and training curve of the Fresca2 dipole. The letters a, b, and c in the quench plot refer to various assemblies and different prestress configurations. [11].

The program also included a task dedicated to the design and construction of an HTS race track with no bore. The EuCARD HTS racetrack was conceived as a basic R&D exercise on HTS for the European accelerator magnet community, and its main features are:

1.  The magnet is a stack of flat coils in racetrack shape—i.e., the coil end is not flared, leaving no access to the coil midplane (no bore). The coils are wound as flat pancakes. Three pancakes, or coil layers, are above the midplane and three exactly symmetrical ones are in the bottom coil.

2.  The conductor is composed of two REBCO tapes, 12 mm wide, soldered face to face to form a pair that sandwich a pure-Cu ribbon in the center (for a total of 70 μm of copper in the sandwich when the few μm of copper coating on the side (see Figure 3 left are included). The stack is about 200 μm thick and a 130 μm Cu-Be ribbon is soldered on each side, so the total conductor thickness is about 460 μm. The conductor is then insulated with polyester film. Finally, two conductor units are co-wound to form a high current cable. A schematic of the cable is shown in Figure 3.

3.  The cable described above is not transposed, and the inner conductor of the cable has smaller inductance than the outer one. To compensate for this effect, each pancake on the top part (wrt to the midplane) is connected to its bottom symmetrical companion in such a way that the current in the inner conductor is in series with the outer conductor of the same cable of the bottom pancake. The coil layout and magnet mechanics are depicted in Figure 4. The magnet is first assembled for a stand-alone test with a demountable structure, where the force and prestress can be adjusted. The structure is also easily demountable to allow for various assembly trials. For the final configuration, as a high-field insert inside Fresca2 a more compact structure is necessary to keep the e.m. forces; see Figure 4.

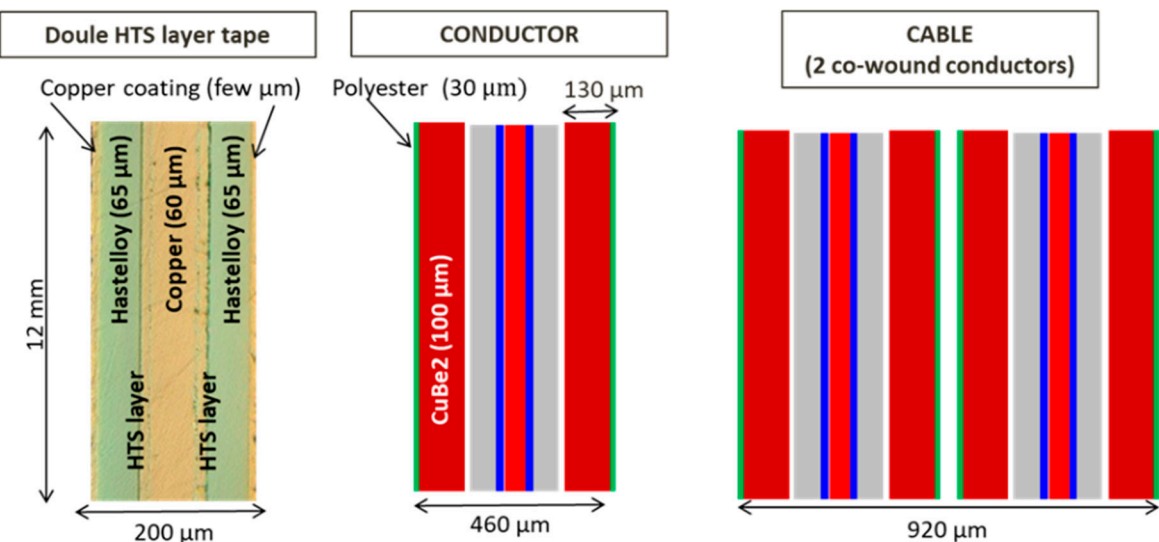

**Figure 3.** Left: picture of the cross-section of the basic HTS unit (two HTS tapes soldered to a central Cu ribbon); center: scheme of the conductor, with the double tape in between two Cu-Be ribbons with insulation; right: the cable composed of two independent conductors (courtesy of A. Ballarino, CERN and M Durante, CEA).

The project has been relatively long. Launched initially in 2008 as part of FP7-EuCARD, it suffered from a lack of resources and was completed in the years 2013–2017 in the frame of the CERN–CEA collaboration. CERN procured the conductor, manufactured by Superpower, Inc. (Glenville , NY, USA), and CEA (Saclay, F) was in charge of the magnet design, fabrication, and testing, with some financial support from CERN. The magnet after a few tests successfully reached 5.4 T in stand-alone, as measured with Hall probes in the magnet center; see the schematic cross-section in the middle position of Figure 4,

which is consistent with the design (it originally was 6 T but then the number of the turns was reduced).

Now it is being prepared for testing in the Fresca2 facility at CERN, and, despite delays due to the COVID-19 pandemic, a test is foreseen by 2021.

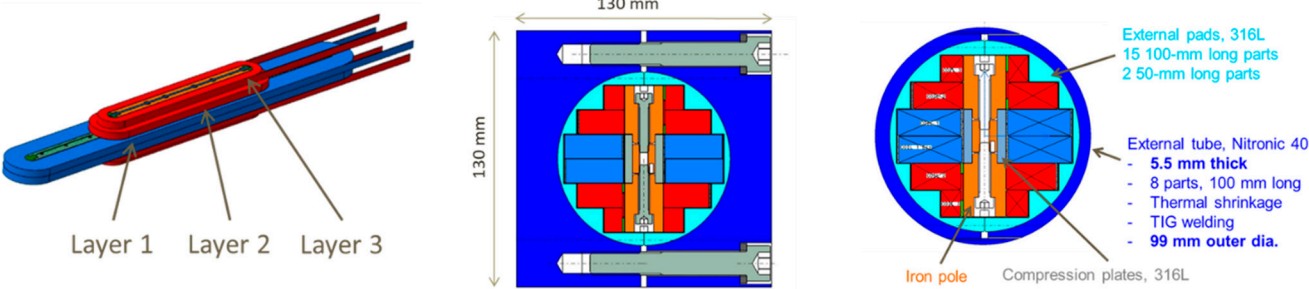

**Figure 4.** The layout of the EuCARD HTS magnet by CEA. Left: view of the three coil layers. Center: first demountable assembly for the stand-alone test. Right: compact assembly for the test in Fresca2. Courtesy of M. Durante, CEA.

## 4. The FP7-EuCARD2 and H2020-ARIES Programs: Overview

Despite the fact that in 2012–2013 the EuCARD HTS magnet was far from completion, in view of the potential use of HTS for any post-LHC collider a program called EuCARD2 (European Collaborative Accelerator R&D-2) was proposed to the European Commission framework programme7, EC-FP7, in 2011. The basic ideas are outlined in [12]. The program was then approved with 80% financing in 2012 and finally started in 2013. The effort on HTS conductor and magnets was named *WP10-Future Magnets* and was part (about 15%) of the much larger EuCARD2 program. Led by CERN and including most institutions active in accelerator magnets in Europe, as well as a few other laboratories, the EuCARD2 WP10 future magnets program [13] had the main goals of manufacturing and qualifying in real demonstrator coils and magnets an HTS cable with characteristics useful for accelerator magnets.

The EU-funded program lasted four years, ending in April 2017; however, since the beginning all work was managed as a long-term collaboration, for most partners extending two or three years more, to properly complete the first R&D phase. The main tasks were conductor development (tapes, cable, characterization), magnet development (design, technology, construction), and stand-alone tests of the coil/magnets in a suitable facility capable of providing a variable temperature and high current. The test in a background field (Fresca2) was left out of the EU program, to be pursued later by CERN and CEA. However, the test in a high-background field was actually dictating the whole design of the magnet.

EC-FP7 provided about 1.3 M€—i.e., about 40% of the initial 3.3 M€ direct cost of the program. The actual direct cost (without overheads) was in total 5.8 M€: the 1.2 M€ additional funding for more conductor and coils was added by the institutes (mainly CERN). Eleven institutes collaborated in the EuCARD2-Future Magnets program:

1. CERN, Geneva, CH, (general coordinator and participating in all tasks and, in particular, responsible for the design and construction of one type of magnet and magnet testing);
2. CEA-Saclay, FR, (in charge of coordinating magnet design and responsible for the design and construction of one type of magnet);
3. Bruker HTS (BHTS), Alzenau, DE, the industry in charge of developing and manufacturing the REBCO tapes (including re-coating after tape punching);
4. KIT (Karlsruhe Institute of Technology), DE, in charge of producing the Roebel cable (punching of the tapes and assembly in the cable of meandered tapes);

5. University of Geneva (CH), University of Twente (NL), and University of Southampton (UK), in charge of the various characterizations of the conductor, both in the form of tape and cable;
6. INFN-LASA (Milan branch of the Italian Institute for Nuclear Physics), in charge of preparing a test station and carrying out one test;
7. Tampere University in Finland, in charge of quench protection simulation;
8. INP of Grenoble (FR), in charge of special magnet design;
9. DTI, the Danish Institute of Technology, giving support to the magnet design and construction.

One of the most successful initiatives of the collaboration was the organization of a series of workshops, WAMHTS—i.e., workshop on accelerator magnets in HTS. The workshops are open to additional laboratories or universities, both in Europe and all over the world, especially the US and Japan. In many of these workshops, we had the participation of industry. The first workshop was held in Hamburg in May 2014, in the frame of the EuCARD2 program, and the series is now under the coverage of successor program, called ARIES (see Section 4.2). The need to share experience in detail and mutualize the resources makes these workshops very useful. The workshops were attended by 50–80 people and the last one, the fifth of the series, was held in Budapest on 11–12 April 2019.

*4.1. Objectives of EuCARD2.*

A high-$J_c$ material is the necessary, however not sufficient, condition to enable the performance of magnets. Accelerator magnets are demanding, needing:

- A high current density in the coil package (typically 400 A/mm$^2$) at the relevant field in order to make magnets of a reasonable size and affordable cost. For our case, this translates to the following requirements:
    - High current density over the whole tape or wire cross-section, called $J_{engineering}$. Our goal was $J_e > 400$ A/mm$^2$@20 T and 4.2 K (taken as a reference operating temperature), with the magnetic field perpendicular to the broad face of the tape (worst direction).
    - A compact cable with a high filling factor to avoid the excessive dilution of the current density in the coil package.
    - Thin and robust insulation.
- Operating current in the conductor in the 5–20 kA range—i.e., a 10 kA-class cable.
- Multi-strand conductor with strand transposition and contact resistance among strands low enough to enable current transfer but high enough to avoid field-quality disruption during ramp. Strands are flat REBCO tape rather than round wire, as in the LTS cables.
- Control of field quality within a few units (one unit being 100 ppm of the main field). In this initial R&D phase, a few tens of units are considered sufficient both for magnetization and for winding geometry.

The scope of the program could be condensed into two main objectives:

1. Produce a 10 kA-class conductor of at least 20–30 m unit length (requiring in total approximately 1 km of 12 mm-wide tape). For this, we set goals for a tape of $J_e = 400$–600 A/mm$^2$ at 20 T, 4.2 K.
2. Build various small dipole magnets with some accelerator characteristics (accessible bore for beam bending, high-order harmonics of the order of 0.1%) to qualify the conductor in near-to-operating conditions.

Besides gaining experience with HTS technology, we were learning how to protect a HTS magnet with a high current density (we want to operate at 4.2 K), qualifying new quench detection technologies, and dealing with screening effects for field quality and quench. Finally, an important objective was learning how to test high current density HTS magnets.

The conductor is fully described in Section 5. Here, it is important to underline that the critical choice was selecting the type of conductor to use; see ref. [12], where the possibility of using Bi-2212 was initially envisaged. The selection was made at the very beginning of the project based on various considerations, including magnet design and construction. The first choice was to select the type of superconductor: REBCO tape was preferred to Bi-2212 round wire, and the reasons for this are explained in an official EuCARD2 document [14], mainly driven by magnet construction considerations and the diversification of sources. Roebel cable was chosen for its high filling factor (similar to the classical Rutherford cable) and because transposition was considered a key feature.

The magnet was fixed with a free bore of 30–40 mm and 200–300 mm of straight section. Three types of magnets were pursued, as detailed in Section 6:

1. A coil block design, called Aligned Blocks or AB dipole—i.e., with Roebel cable positioned such that the tape broad face was parallel to field lines for maximizing the current density) and flared ends. This design was pursued mainly by CERN.
2. A classical cosθ design to explore synergy with a design commonly used in the accelerator domain. This design was taken up by CEA.
3. A coil block design based on stacked tape (rather than Roebel cable similar to the previous two layouts). The design was proposed and developed by INPG (Institut d'Ingégnerie, Grenoble, Fr).

Two full dipoles have been engineered, built, and tested in the AB layout, called FeatherM2 dipoles, and a third one is still being manufactured at present. One dipole in the Cosθ layout has been manufactured and is in the final assembly stage for testing. The third design, with a stacked table conductor, did not go beyond the conceptual design level.

### 4.2. Objectives of ARIES

At the end of 2015, well before EuCARD2 was finished in 2017, the successive European program, ARIES, was determined. Since the magnet community was very busy with HiLumi LHC and the scarce R&D resources were still engaged with the new high-field $Nb_3Sn$ program for FCC and with the completion of EuCARD2, there was little room for a new initiative concerning HTS for accelerators in Europe. Therefore, we could obtaining only a small amount of resources, about 500 k€, that were devoted to one scope: improve the engineering current density of the REBCO tape produced for EuCARD2 by Bruker. ARIES, being a pure conductor development program only, is detailed in Section 5.

### 5. The Conductor R&D Programs in EuCARD2 and ARIES

Even if HTS conductors have been used to generate high fields in solenoids for many years [15–18], their validation as technical conductors suitable for accelerator magnets in the 20 T range is in its infancy. As previously mentioned in Section 4, one of the main requirements for accelerator magnets is a high engineering current density in the basic conductor element (tape in our case), indicated as $J_e$ and defined as the ratio of the critical current to the total conductor cross-section. The performance target for $J_e$ at the operating field and temperature is in the order of 500 A/mm$^2$. Only two industrial HTS conductors have achieved the desired $J_e$ performance in the 20 T range: $YBa_2Cu_3O_{7-x}$ (YBCO or REBCO, used to indicate the use of a Rare Earth (RE) element other than Yttrium, Y) and $Bi_2Sr_2CaCu_2O_{8+x}$ (BSCCO-2212). REBCO comes in the form of a thin tape ready for use, where the thin layer of Rare Earth superconductor (few μm thickness) is deposited on a thin (tens of nm to few μm) multifunctional oxide barrier, typically made of an $Al_2O_3$ diffusion-barrier layer, an $Y_2O_3$ or $CeO_2$ seed layer, and a biaxially textured MgO or yttrium-stabilized zirconia (YSZ) layer, which acts as a buffer between REBCO and a strong substrate (Hastelloy, stainless steel, or NiW) 30 to 100 μm thick. The tape is then coated with a few μm of silver to protect the superconductor and finally some copper (5–100 μm) is added for stabilization either via electrodeposition or via soft soldering. Industrial BSCCO-2212 is produced as a round wire, using procedures and tooling that closely resemble those used for the production of LTS wires. BSCCO-2212 has recently been demonstrated to have

record $J_e$ values close to 1000 A/mm$^2$ at 4.2 K and 30 T [19]. However, the necessity to perform a high-temperature heat treatment at high pressure (up to 100 bar) after winding in order to reach the best performance represents a complication to its penetration in the applications. A strategic decision on conductors taken in the three above-mentioned European programs for accelerator magnets was to focus almost exclusively on one of the two materials—i.e., REBCO—and thus avoid excessive diversification. This choice, mainly driven by technical considerations of the better mechanical properties and simpler magnet technology (see [14]), also provided complementarity among the EU and US-based programs, where BSCCO-2212 was supported by the US-DOE Conductor Development Program [20] and the Bismuth Strand and Cable Collaboration (BSCCO) [21]. After the high $J_e$, another important requirement for accelerator magnets is small inductance for use in long magnet strings; hence, the conductor in the winding must have a large current carrying capacity in the range of 10 kA. As the current carrying capability of BSCCO-2212 wires and REBCO tapes at the envisaged operating conditions is in the 100–1000 A range, it is imperative to use a compact multi-strand cable. While the choice of the Rutherford cable configuration is straightforward in the case of BSCCO-2212 round wire, it is not compatible with REBCO thin tapes. However, alternative geometries appear to be suitable for the high current density required by accelerator magnets: stacked tapes in various configurations [22–24], Conductor on Round Core (CORC$^®$) [25], and Roebel Cable [26] (see Figure 5). Special stacked tape cables were used for the EuCARD race-track insert [27], while in EuCARD2 the Roebel configuration was preferred after a review of the various cable options [28].

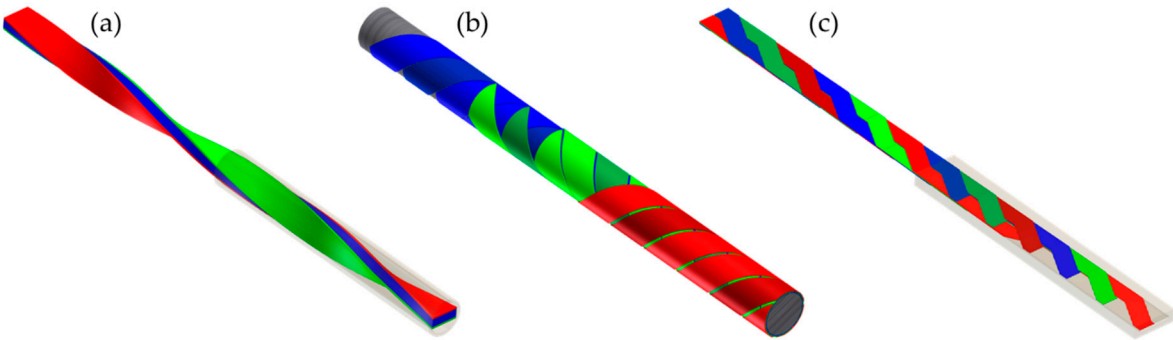

**Figure 5.** Schematic illustration of different REBCO-based cables: a twisted stacked tape cable [22] (**a**), a Conductor on Round Core (CORC$^®$) [25] (**b**), and a Roebel Cable [26] (**c**). Adapted from Ref. [28].

In contrast with the EuCARD activity on HTS, which contained only a task on the design and construction of an insert, EuCARD2 focused on a strong HTS conductor program linked to a magnet R&D part, the goal being the creation of a bridge between materials and large-scale magnets. REBCO tapes were mainly provided by Bruker HTS (BHTS), a member of the EuCARD2 consortium.

While the development of other tape manufacturers was initially driven by the perspectives of applications in the electrical utility sector, the R&D program of BHTS was dedicated from its beginning to developing and producing long-length tapes for high-resolution NMR spectrometers [29], whose targets in terms of high current density at high field and low temperature have considerable synergy with the requirements for accelerator magnets. Table 1 summarizes the targets and the minimum required performance for the REBCO tapes considered within the scope of EuCARD2 and ARIES. The table includes the engineering current density at T = 4.2 K and B = 20 T, $J_e$ (4.2 K, 20 T); the critical current homogeneity over the length, $\sigma(I_c)$; the magnetization at B = 1 T and T = 4.2 K, $\mu_0 \Delta M$ (1 T, 4.2 K); the allowable transverse stress, $\sigma_{transverse}$; the longitudinal strain, $\varepsilon_{longitudinal}$; and the tape unit length.

**Table 1.** Targets and minimum required performance for the REBCO tapes in EuCARD2 and ARIES.

| Parameter | EuCARD2 | | ARIES | |
|---|---|---|---|---|
| | Ultimate Target | Minimum | Ultimate Target | Minimum |
| $J_e$(4.2 K, 20 T) (A/mm$^2$) | 600 | 400 | 1000 | 600 |
| $\sigma(I_c)$ (%) | $\leq 10$ | | | |
| $\mu_0 \Delta M$ (1 T, 4.2 K) (mT) | $\leq 300$ | | | |
| Allowable $\sigma_{transverse}$ (MPa) | | >100 | | |
| Allowable $\varepsilon_{longitudinal}$ (%) | | >±0.3 | | |
| Unit Length (m) | $\geq 100$ | >50 | | |

EuCARD2 set a $J_e$ ultimate target of 600 A/mm$^2$ at 4.2 K and 20 T in a perpendicular field orientation, with a minimum $J_e$ of 400 A/mm$^2$ to be considered for magnet design. The goal of ARIES was to further increase $J_e$, aiming at a minimum of 600 A/mm$^2$ and finally targeting 1000 A/mm$^2$. These being the first R&D activities on HTS conductors for accelerator magnets, it was decided to reduce demands for the homogeneity of the critical current in a unit length and for the magnetization amplitude. Both values reported in Table 1 are more than ten times larger than those achieved in the production of Nb-Ti for LHC and about three times larger than the specifications for the Nb$_3$Sn wires of HiLumi. As for the mechanical characteristics, a maximum transverse compressive stress of 100 MPa and a longitudinal applied strain of ±0.3% were specified, under which the superconductor should retain mechanical integrity and a minimum of 90% of its virgin $J_e$ (before loading). The levels above are the minimum requirements for a high-field accelerator magnet, and are not particularly challenging for the REBCO tapes, whose metallic substrate is typically a hard metal with a high elastic modulus and large elastic strain limit. Moreover, REBCO tapes typically exhibit an out-of-plane bending radius below 10 mm and a sufficiently large in-plane bending radius in the order of 1 m, making it possible to accommodate coil ends with an aperture. The last target concerns the unit length (UL) of the superconductor. A minimum length of the order of 50 m was required, which was largely in excess of the length of a pole in both the aligned blocks and the cosθ designs. In the end, piece lengths of 30 m were also accepted (minimum length to wind a pole was 26 m).

*5.1. Progress in the Performance of REBCO Tapes: Fabrication*

The development of practical superconductors based on REBCO relies on two common features: a bi-axially textured template consisting of a flexible metallic tape (Hastelloy, stainless steel, or NiW) coated with a multifunctional oxide barrier, and an epitaxial REBCO layer. The textured template, which is needed to eliminate all but low-angle grain boundaries in REBCO, is created by either deforming the metal substrate with rolling-assisted bi-axially textured substrate technology (RABiTS) [30] or by texturing the buffer layers deposited on the metal substrate by so-called ion beam-assisted deposition (IBAD) [31] and its variant, alternating beam-assisted deposition (ABAD) [32]. The epitaxial REBCO layer is grown either by physical routes, such as pulsed-laser deposition (PLD) [33–35] and reactive co-evaporation (RCE) [36,37], or by chemical routes, such as metal organic deposition (MOD) [38] and metal organic chemical vapor deposition (MOCVD) [39].

BHTS was the partner responsible for manufacturing the tape in both EU-funded programs, EuCARD2 and ARIES. CERN procured additional 500 m tapes with its own funding with the same specifications as those of EuCARD2, doubling the quantity manufactured by BHTS to give some continuity to the production line and making it possible to manufacture more coils. The technological chain of BHTS was established for the deposition of YBCO on Cr-Ni stainless steel tapes with widths of 4 or 12 mm. ABAD is the proprietary process of BHTS employed for the deposition of the yttria-stabilized zirconia (YSZ) buffer layer, typically 2–3 μm thick, which serves as a template for the bi-axially textured growth of YBCO. The YSZ layer thus has an equivalent function to the MgO layer resulting from the IBAD process used by other manufacturers, but it is approximately 5 to 10 times thicker. The YBCO layer, whose typical thickness lies in the 1.5–2 μm range, is deposited on the

buffered stainless steel tape by PLD using drum-based tape transport. The tape is wound on a tubular drum and transported through a deposition zone without any rewinding. For the EuCARD2 and ARIES productions, a new machine called PLD-300 that was jointly owned by BHTS and CERN was set up for this process: its drum is capable of producing unit lengths of about 300 m of 4 mm-wide tape or, alternatively, 100 m of 12 mm-wide tape. While an obvious limitation of length is present, this technique provides the benefit of avoiding the contamination of the tape by friction, which may be generated in a reel-to-reel deposition process.

The production process of BHTS was fine-tuned for deposition on a 100 μm-thick tape. This relatively high thickness enables favorable mechanical properties that are important for ultra-high-field solenoidal magnets but is deleterious for the $J_e$ performance. The value of $J_e$ at 4.2 K and 20 T for the material produced at the beginning of the EuCARD2 collaboration in 2013 was below 300 A/mm$^2$ (see Figure 6). Given the layered structure of the REBCO tapes, as shown in Figure 7, there were three possible strategies to enhance the $J_e$:

(i)    Increasing the thickness of the superconducting layer;
(ii)   Increasing the critical current density, $J_c$, of the superconducting layer by introducing artificial pinning centers;
(iii)  Reduction in the substrate thickness.

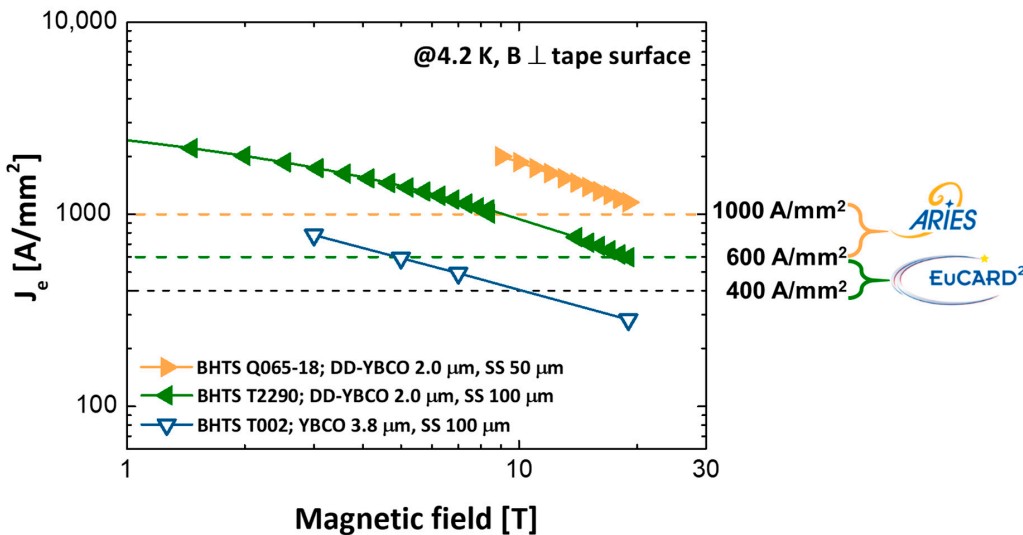

**Figure 6.** Evolution of the engineering critical current performance of YBCO tapes from BHTS and EuCARD2 and ARIES $J_e$ targets. $J_e(B)$ is measured at 4.2 K in a perpendicular orientation for three tapes: T002, produced at the beginning of EuCARD2; T2290, produced during EuCARD2; and Q065-18, produced during ARIES.

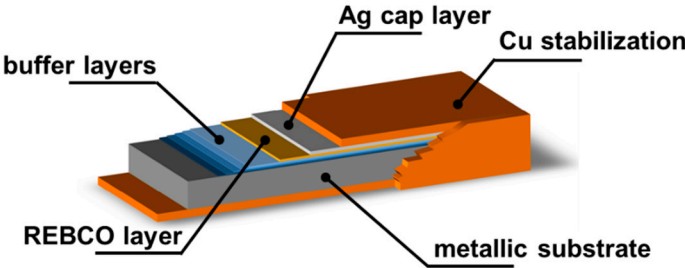

**Figure 7.** Schematic view of the layered structure of a REBCO tape.

The growth of thick REBCO films appears to be a straightforward opportunity to improve $J_e$. However, common to most REBCO film growth techniques, the superconducting critical current density $J_c$ was typically found to strongly deteriorate with the thickness

of the REBCO layer, $t_{REBCO}$, when $t_{REBCO} \geq 3$ μm [40–42]. As $J_e = J_c \cdot t_{REBCO}/t_{tape}$, where $t_{tape}$ is the total thickness of the tape, the increase in $J_e$ is jeopardized by the competition between the increase in $t_{REBCO}$ and the resulting degradation of $J_c$. Significant research effort has been devoted worldwide towards improving the in-field $J_c$ by introducing nanoscale defects—e.g., $BaZrO_3$ (BZO) [43,44] and $BaHfO_3$ (BHO) [45]—in the REBCO matrix. In-field $J_c$ depends primarily on the angle between the field direction and the REBCO crystallographic orientation: anisotropy makes $J_c$ maximum when the field is parallel to the tape surface and minimum when it is perpendicular to the tape surface. Moreover, the ratio of the current densities in the two orientations tends to increase with an increasing magnetic field. By itself, the anisotropy is not detrimental but it introduces an extra layer of complexity in the magnet design process. In a magnet, the conductor is exposed to a magnetic field whose angle depends on its position in the winding. This implies that the load line at each point of a REBCO-based coil virtually intercepts a different $J_c(B)$ curve. A careful adaptation of the magnet design process to the specific anisotropic tape characteristic is necessary in order to determine the safety margins of each design. Depending on the technique adopted for the REBCO film growth, the morphology and size of the artificial nanoscale defects and thus their effects on the pinning properties are different. In particular, PLD and MOCVD lead to the formation of anisotropic nanorods (~5 nm in diameter, >100 nm in length) of BZO or BHO that grow along the c-axis [46,47]. The intrinsic weak pinning in the magnetic field parallel to the c-axis—i.e., perpendicular to the tape surface—is thus reinforced by the interaction between nanorods and vortices.

Substantial progress was made in this direction by BHTS. A significant increase in $J_c$ in a high magnetic field was recorded in tapes manufactured employing the so-called double-disorder (DD) route [29,48,49], which creates intrinsic and extrinsic defects in the YBCO layer. The intrinsic disorder is caused by local stoichiometry deviations with respect to the nominal $YBa_2Cu_3O_{7-x}$ composition and is achieved through a tailored modulation of the oxygen pressure during the PLD process. The $O_2$ pressure variation causes a change in the mean-free path for the different atoms in the plasma plume generated by the laser ablation of the YBCO target. The result is the formation of different defects of crystallinity (e.g., stacking faults, nano islands, nano-precipitates, etc.) in the YBCO film that serve as pinning centers. The extrinsic disorder is obtained by the 5 wt% content of BZO in the YBCO target, resulting in the appearance of local and organized BZO nanorods mostly oriented along the c-axis of the YBCO film. A total length of more than 600 m of YBCO tape was produced by BHTS in the frame of EuCARD2, to which 500 m of CERN-procured post-EuCARD2 length should be added, in lengths from short samples (order of 1 m) to widths and lengths that were relevant to the conductor needs for the dipole demonstrators (the minimum required length of 12 mm tape was about 30 m and the longest one delivered with homogeneous properties was about 90 m). As shown in Figure 6, the target $J_e$ of 600 A/mm$^2$ at 4.2 K, 20 T, set at the beginning of the EuCARD2 activities was achieved in the perpendicular field orientation—i.e., when the $J_c$ is at its minimum. Thanks to the optimization of the DD route, the latest tapes systematically exceeded the target value, with the highest $J_e$ extrapolating towards 900 A/mm$^2$ at 4.2 K, 20 T [48].

Based on the technical success of EuCARD2, the primary goals of ARIES were set to further increase the target $J_e$ by a factor of 1.5–2 and to industrialize the process for long lengths (>25 m). The most straightforward way to reach the desired performance appeared to be a reduction in the stainless steel substrate thickness from 100 to 50 μm. Nonetheless, the implementation of this strategy proved to be harder than expected. Major core steps in the ABAD and PLD processing routes had to be retuned because of a substantial variation in the mechanical and thermal parameters of the thinner substrate [49]. In particular, the deposition of the relatively thick YSZ buffer layer on the 50 μm substrate determined a pronounced transverse bending of the tapes in a bow shape that was traced back to the thermal contraction mismatch between the ceramic layer and the stainless steel (see Figure 8). Even though this effect does not seem to have a great influence on the critical current, it can be detrimental for tape manipulation during cabling and winding.

In particular, the mechanical positioning for tape cutting—i.e., the meandering formation during Roebel cable formation described in Section 5.3.—certainly becomes more difficult. To mitigate the bow shape, BHTS also started depositing the YSZ film on the back of the tape. As a result, the tape has a significantly lower bowing height, as shown in the tape on the left side of Figure 8. Despite all these challenges, BHTS has been able to produce tapes with a very high critical current density. The Cu stabilizer has been tailored to obtain a good $J_e$ overall, with a mix of standard 20 $\mu$m Cu thickness per side (i.e., 2 $\times$ 20 $\mu$m total Cu thickness) and 7 $\mu$m Cu thickness per side (i.e., 2 $\times$ 7 $\mu$m total Cu thickness). Several units between 15 m and 75 m reached or exceeded the minimum $J_e$ (4.2 K, 20 T) of 600 A/mm$^2$. The best unit, a 12 mm-wide tape that was originally 25 m long and prepared with 2 $\times$ 7 $\mu$m Cu stabilizer, achieved a $J_e$ of 1153 A/mm$^2$ at 4.2 K, 19 T, in a perpendicular field orientation (see Figure 6). Figure 9 reports the magnetic field dependence of $J_e$ in the perpendicular orientation at various temperatures between 4.2 K and 40 K for an equivalent tape prepared with a thicker 2 $\times$ 20 $\mu$m Cu stabilizer: the value of $J_e$ (20 K, 19 T) is 430 A/mm$^2$—i.e., it exceeds at 20 K the minimum value set for $J_e$ at 4.2 K in EuCARD2. This gives evidence of the substantial impact of the developments driven by accelerator magnet R&D to the recent progress in the technology of long REBCO tapes.

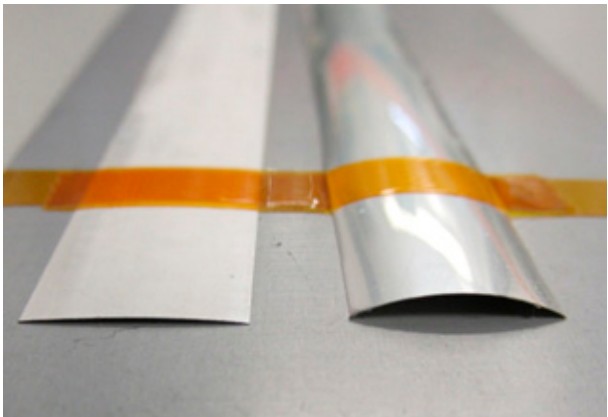

**Figure 8.** Bow shape of the YBCO tapes deposited in the 12 mm-wide 50 $\mu$m-thick stainless steel substrate. The effect is mitigated when the YSZ buffer layer is deposited on the two faces of the substrate (tape on the left). Figure adapted from Ref. [35].

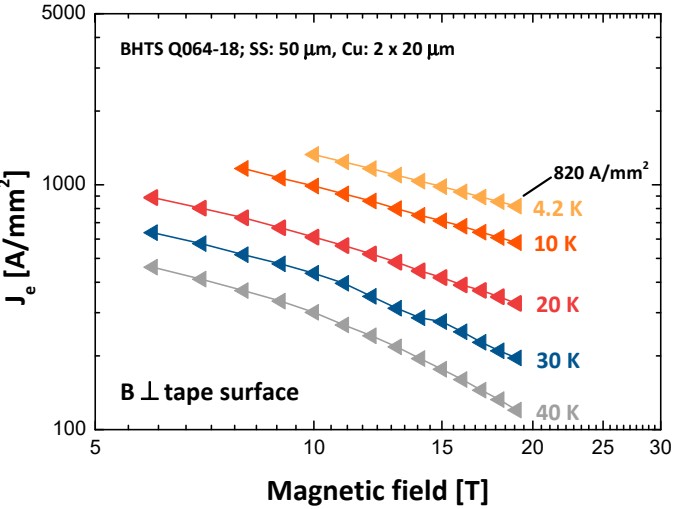

**Figure 9.** Magnetic field dependence of $J_e$, in the perpendicular orientation, at various temperatures between 4.2 K and 40 K for a 12 mm-wide BHTS tape with a 50 $\mu$m-thick stainless steel substrate and 20 $\mu$m Cu stabilizer on each side.

*5.2. Progress in the Performance of REBCO Tapes: Characterization*

The high $J_e$ in the high magnetic fields of REBCO tapes is a necessary but not sufficient condition from the perspective of their use in accelerator magnets. To qualify REBCO tapes for operation in very high-field coils, magnet design relies on the control of the electrical, mechanical, and thermal properties of the superconductor. The performance in these respects is strongly influenced by the fabrication route, tape layout, and materials used, and these parameters vary from one manufacturer to another. It has already been pointed out how a detailed knowledge of the anisotropy of the critical current, $I_c$, at the operating temperature and field is important for magnet design. Moreover, the characterization of the temperature dependence of $I_c$ between the operating temperature and the current-sharing temperature, $T_{cs}$, is necessary to consolidate the electromagnetic models simulating the coil behavior in case of quenches. Concerning the quench protection, the operating temperature margin is significantly larger compared to LTS-based magnets. This results in a much slower normal zone propagation velocity, making quench detection difficult. The management of the mechanical loads imposed by the electromagnetic forces during operation is another critical issue.

In the frame of EuCARD2, the Group of Applied Superconductivity at the University of Geneva performed an extensive measurement campaign to compare the electromagnetic, electromechanical, and thermophysical properties of tapes from various industrial manufacturers, and a detailed analysis is reported in Refs. [50–54]. The results obtained for tapes from American Superconductor, BHTS, Fujikura, SuNAM, SuperOx, and SuperPower are summarized as radar charts in Figure 10, which provides a snapshot of the performance of industrial tapes in 2015, at the time of the project. The axes of the radar charts report $J_e$ (4.2 K, 19 T) and $J_c$ (4.2 K, 19 T) in perpendicular field orientation; the $J_e$ (77 K, self-field), the residual resistivity ratio (RRR), and the area fraction of the Cu stabilizer; the total cross-section area of the tape; the irreversible strain limit, $\varepsilon_{irr}$; and the irreversible stress limit, $\sigma_{irr}$. Some of these properties are discussed in more detail in the following text. However, Figure 10 gives an instant picture of the large spread of performance among manufacturers.

Figure 11 shows the relation between longitudinal strain and mechanical stress at 4.2 K for the various tapes. Young's moduli are in the 155–187 GPa range, highest for BHTS and lowest for SuperPower [51]. All the curves are nonlinear even at low strains due to the presence of the Cu stabilizer, which quickly reaches its yield point. The BHTS stress–strain curve is more rounded compared to the other samples because of its stainless steel substrate, which makes it the sample with the lowest yield strength, 736 MPa and 499 MPa for the tapes with substrates of 100 μm and 50 μm, respectively. The highest yield strength is observed for the REBCO tape from SuperOx, about 1000 MPa, and this is mainly due to its strong Hastelloy substrate and to the low Cu cross-section area. From the diverging stress–strain behavior, one would expect these tapes to also differ significantly in their electromechanical properties. However, there are no differences in the longitudinal stress dependencies of the critical current below 600 MPa, as shown in Figure 12 at 4.2 K, 19 T. The irreversible stress limits of all the examined samples are in the 740–840 MPa range [51]. Due to its rounded stress–strain relation combined with low yield strength, the BHTS tape with a 100 μm-thick substrate exhibits the lowest irreversible stress limit among those reported in Figure 12. The BHTS tape with 50 μm-thick substrate, produced in the frame of ARIES, has not yet been tested, but it is reasonable to expect a further reduction in the irreversible stress limit due to the lowered yield strength.

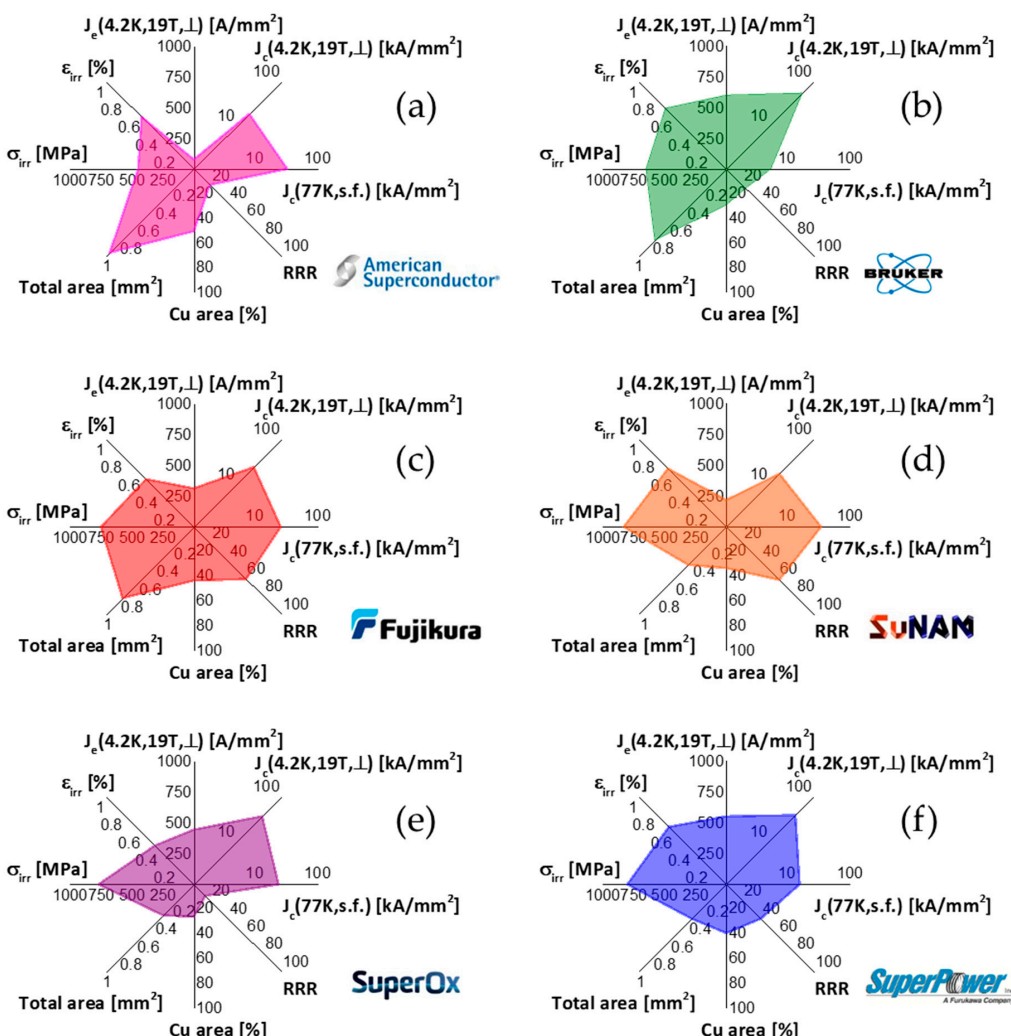

**Figure 10.** Radar charts summarizing the main properties of REBCO tapes from six industrial manufacturers: American Superconductor (**a**), BHTS (**b**), Fujikura (**c**), SuNAM (**d**), SuperOx (**e**), and SuperPower (**f**). The reported parameters compiled from Refs. [36–38] are the engineering current density $J_e$ (4.2 K, 19 T) and the current density in the REBCO layer $J_c$ (4.2 K, 19 T) in the perpendicular field orientation; the engineering current density $J_e$ (77 K, self-field); the residual resistivity ratio (RRR); the area fraction of the Cu stabilizer (Cu area), the total cross-section area of the tape (total area); the irreversible longitudinal strain limit, $\varepsilon_{irr}$; and the irreversible longitudinal stress limit, $\sigma_{irr}$.

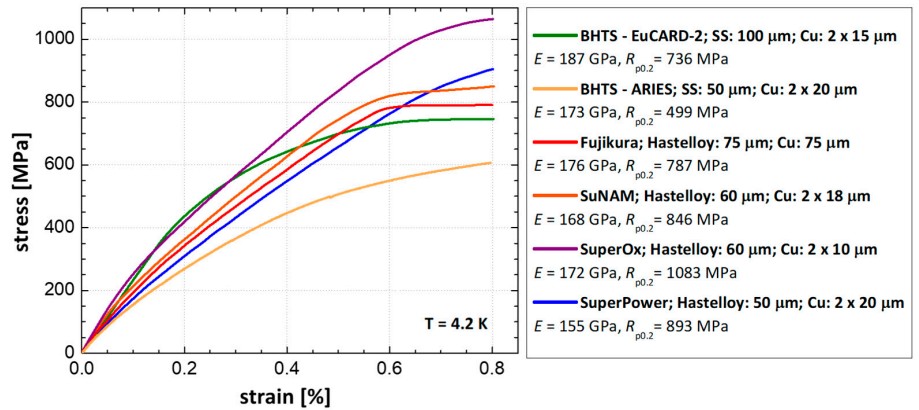

**Figure 11.** The stress–strain dependence of six industrial REBCO tapes is shown at 4.2 K, with the indication of the Young's modulus, E, and yield strength, $R_{p0.2}$, values. Adapted from Ref. [51].

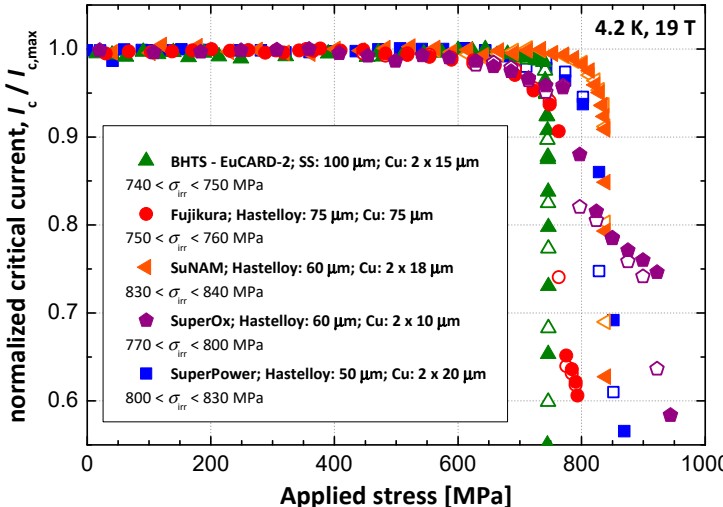

**Figure 12.** Normalized critical current $I_c/I_{c,max}$ versus longitudinal tensile stress, $\sigma$, and irreversible stress limits, $\sigma_{irr}$, at 4.2 K and 19 T for five industrial REBCO tapes. Adapted from Ref. [51].

REBCO tapes are thermally and electrically stabilized with Cu and added to the tape either by co-lamination or by electrodeposition, with the total thickness of the stabilizer ranging between 10 and 100 µm. If local heating occurs during operation, heat and current transfer along the tape through the stabilizer are the main channels for preventing a quench in insulated coils. In the case of non-insulated (NI) windings, conduction in adjacent turns is supposed to become the dominating mechanism. In any case, an adequate amount of Cu is needed, with a sufficiently high RRR. An RRR > 100 is routinely achieved in LTS wires. The Cu stabilizer in the REBCO tapes exhibits significantly lower RRR values ranging between 14 and 61 for the samples reported in Figure 13. In particular, the BHTS tape from one of the EuCARD2 batches had a value of RRR = 23. Through a progressive tailoring of the electrodeposition process, the tapes produced for ARIES regularly reached RRR > 50.

Focusing on the thermophysical properties, the thermal conductivity of the tape, $\kappa$, which is one of the most relevant parameters governing the thermal stability, can be estimated with a reasonably good accuracy in zero field from the RRR and the cross-sectional area of the Cu stabilizer using an analytical formula. The field-induced decrease in $\kappa$ can be qualitatively understood considering the reduction in the electron mean free path of the electrons caused by the action of the Lorenz force and the consequent field-induced reduction in the RRR. The $\kappa(T)$ curves measured at B = 0 T and B = 19 T on the six industrial REBCO tapes are reported in Figure 13a,b, respectively. Comparing the results shown in Figure 13a,b, it follows that the differences among tapes from various manufacturers observed at B = 0 T are strongly reduced after the application of an intense field [52]. This is the consequence of two competing effects: (i) different relative reduction in $\kappa$ on applying the magnetic field; (ii) the different amount of Cu present in the tapes. The field-induced reduction in $\kappa$ in Cu depends on the RRR and samples with higher purity—i.e., Fujikura and SuNAM—show larger variations. On the other hand, since the Cu contribution to the longitudinal thermal conductivity of the tape is proportional to the Cu area, the measured reduction in $\kappa$ also depends on the tape's Cu/non-Cu ratio. It follows that the AMSC tape retains a high $\kappa$ at 19 T in spite of its low RRR thanks to the larger Cu fraction.

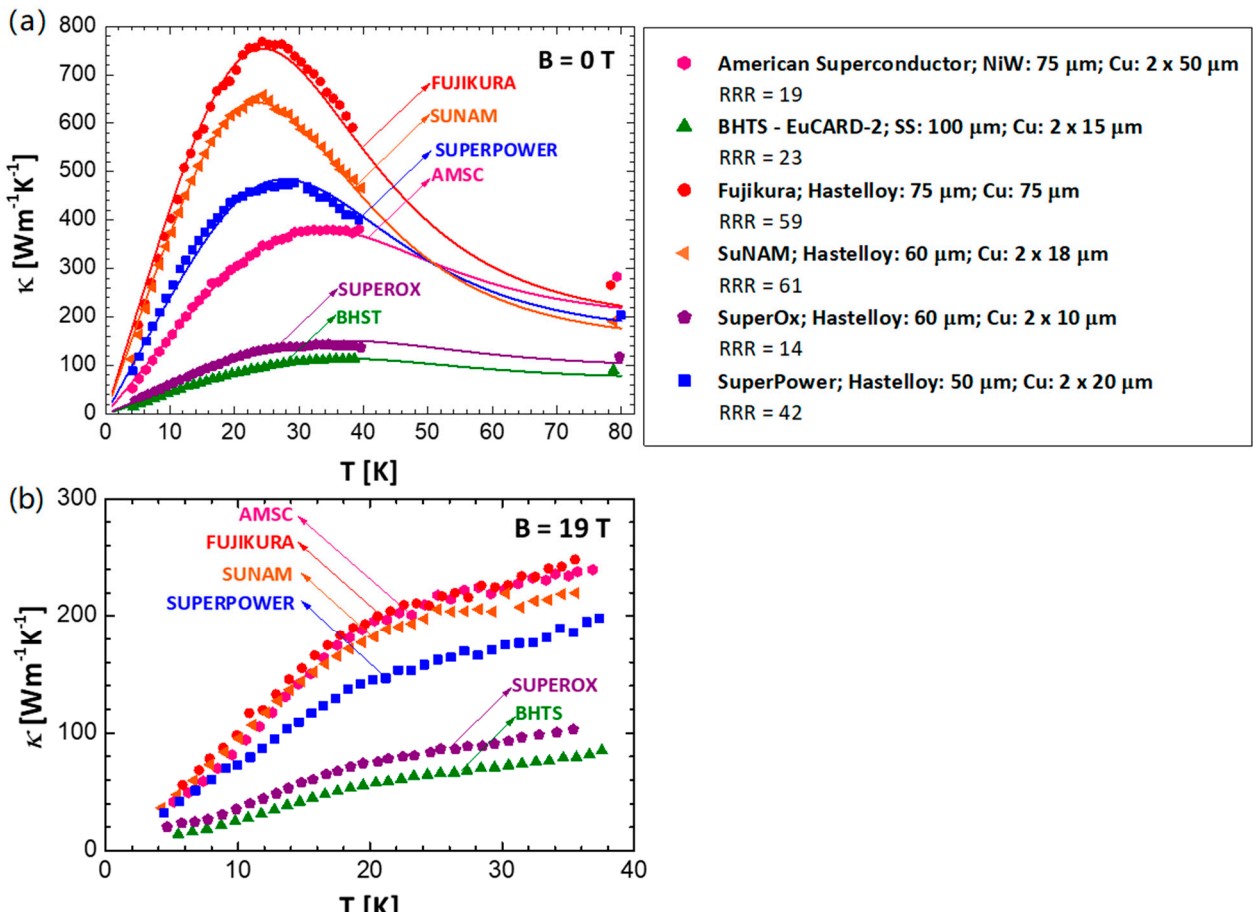

**Figure 13.** Temperature dependence of the thermal conductivity of REBCO tapes from six industrial manufacturers at B = 0 T (**a**) and B = 19 T (**b**). Adapted from Ref. [52].

### 5.3. Development of a 10-kA Class REBCO Cable

5.3.1. REBCO Roebel Cable Design Evolution

The choice of the Roebel cable as a baseline conductor configuration in EuCARD2 [14] was taken based on the criteria of high engineering current density, full transposition, and demonstrated ability to carry currents in the range of 10 kA [55]. This type of conductor was originally conceived by Ludwig Roebel in 1912 as a technical solution to the requirement for low-loss high-current copper cables in conventional electrical machinery and took its name from him [26]. Roebel identified the need for segmenting the conductor into strands insulated from each other and transposing them along the cable direction in order to reduce the induced eddy currents and current loops. Based on the same design concept of its copper counterpart, the first Roebel cable from the REBCO tapes was manufactured at the Karlsruhe Institute of Technology (KIT) in 2006 [56].

A Roebel cable, shown schematically in Figures 5 and 14, is made of meandered-shape punched tapes assembled in a compact configuration. The compaction, defined as the ratio of tape to cable engineering current density, reaches values above 80%, which is similar to the Rutherford cables made out of round wires. In EuCARD2, the Roebel cable was also preferred to other conductor geometries for its full transposition, which is needed to enforce equal current distribution among the tapes. This is a basic paradigm that holds for LTS magnet conductors, and its verification for HTS was one of the tasks of the magnet development activity. The main disadvantage of the Roebel cable is that it requires the punching of the tape, with a loss of costly material and potential degradation associated with the fact that one edge of the tape is cut open. This last issue was solved in EuCARD2

by changing the fabrication sequence—i.e., punching the tapes right after the Ag layer deposition and then coating the resulting meandered tape with Cu. The so-called *punch-and-coat* approach, opposed to the usual *coat-and-punch* sequence, offered the advantage of sealing the slit sides of the tape and thus avoiding exposing the superconducting REBCO layer to air. In a later stage of EuCARD2, this process was further refined by a special Ag coating of the punched surface before coating with copper.

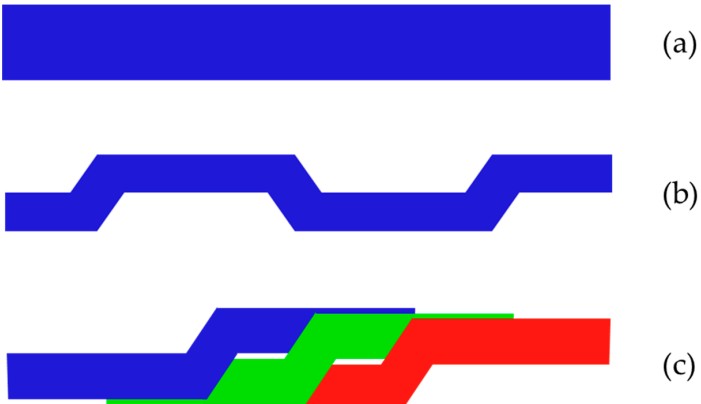

**Figure 14.** Schematic illustration of a full-width tape (**a**), a meandered-shape punched tape (**b**), and a Roebel cable assembly (**c**).

A cable width of 12 mm was chosen for the winding of both magnet variants developed in EuCARD2—i.e., aligned blocks and cosθ (see Section 6). The width of the cable corresponds to the initial width of the tape, which is then cut in meanders with a residual tape width of 5.5 mm. The initial design proposed by KIT, which was the EuCARD2 partner in charge of the manufacture of Roebel cables, accommodated 15 tapes with a 226 mm transposition pitch. However, it turned out to be delicate to use when winding over small radii, as tapes tend to slide differentially in the cable, as well as for the layer jump because the tapes buckle when the cable is bent in the non-easy direction. The experience from winding tests thus triggered an iteration on the cable geometry. This was an important step to include in the cabling and winding validation as well as cables made of tapes that were procured from manufacturers other than BHTS. In practice, two cable geometries were defined to achieve the required critical current target: (i) a cable made of 15 tapes, based on 100 µm-thick tapes (as obtained with a 50 to 60 µm substrate and 40 to 50 µm of Cu stabilizer); (ii) a cable made of tapes of 140 µm thickness (as obtained starting with a 100 µm substrate). These are typically the BHTS tapes produced for EuCARD2 and are described in Section 5.1. Because of the higher $J_e$ compared to other manufacturers, the number of tapes in the cable was reduced to 13, still achieving a sufficiently high current density. For both geometries, the cable transposition pitch was increased to 300 mm to leave some additional space for the slippage of single tapes during winding.

The use of cables with an odd number of tapes in both designs emerged from a study of the effective section that experiences stress under transverse loads [57]. Once wound in a dipole, electromagnetic forces mainly act normal to the broad face of the cable. REBCO tapes have, in principle, an excellent transverse compression strength when the load is applied on the broad face [58]. However, Roebel cables have an inhomogeneous thickness along their length and across their width, which results from the punching procedure, non-uniformities in the tape thickness, as well as from the presence of crossovers. This induces an uneven distribution of the transverse compressive stress applied to the cable and may lead to local stresses in excess of the mechanical limits of REBCO tapes. Moreover, a small transverse effective section may result in large inter-tape contact resistance. Therefore, the management of the transverse effective section is essential for both mechanical and electrical reasons. The 2D geometrical model of Fleiter et al. [57], which determines the relation between the crossing angle of the meander tape, the transposition length, and the

number of tapes, provided the guidelines to optimize the cable design and obtain effective sections above 50% of the total transverse section.

### 5.3.2. Transverse Stress Tolerance of REBCO Roebel Cables

REBCO Roebel cables are being developed in view of 20 T class accelerator magnets. The electromagnetic forces generated in such magnets will lead to sizeable transverse stress on cables, which in the present designs is of the order of 150 MPa [59]. Due to the reduced effective surface, the critical current of bare Roebel cables already starts to degrade at stress levels as low as 40 MPa [58,60]. However, proper impregnation with epoxy resin can dramatically improve the transverse pressure tolerance, helping the stress redistribution by increasing the effective section supporting the pressure.

At the University of Twente, Roebel cables with an architecture directly relevant for the EuCARD2 magnet program were investigated with a variable transverse mechanical load at 4.2 K in a 10.5 T perpendicular magnetic field [61]. Two alternative impregnation methods were tested: (i) CTD-101K, which is used at CERN for the impregnation of large-scale magnets; and (ii) Araldite CY5538 mixed with FW600 EST fused silica powder, following a recipe from KIT [61]. Figure 15 reports the results of the experiments performed on three cables. All the cables were manufactured at KIT and comprise 15 tapes with a transposition length of 226 mm. Cable #1 and Cable #2 were assembled from SuperPower tape and impregnated with Araldite CY5538 and CDT-101K, respectively. Cable #3 was made with BHTS tape and impregnated with CDT-101K. The critical current at 4.2 K, 10.5 T of the Roebel cable made with BHTS tape is 7.75 kA—i.e., 2.5 to 3 times higher than that of the cables made with SuperPower tapes with a similar architecture. This is a consequence of the tailored developments of BHTS to enhance the performance of the tapes at low temperature, high field. All the impregnated cables exhibit a remarkable tolerance to transverse stress and satisfy by far the design requirements of the presently envisaged HTS accelerator magnet demonstrators: no critical current degradation is observed up to 440 MPa in Cable #1 and Cable #2, and up to 370 MPa in Cable #3. Only at higher stress levels, the cables show a gradual but irreversible reduction in critical current. The cross-sections of the cables were examined by optical microscopy to check for visible damage of the tapes due to the transverse loading. Interestingly, Cable #3, which was made with the *punch-and-coat* approach, is the only one that does not exhibit signs of tape delamination [61].

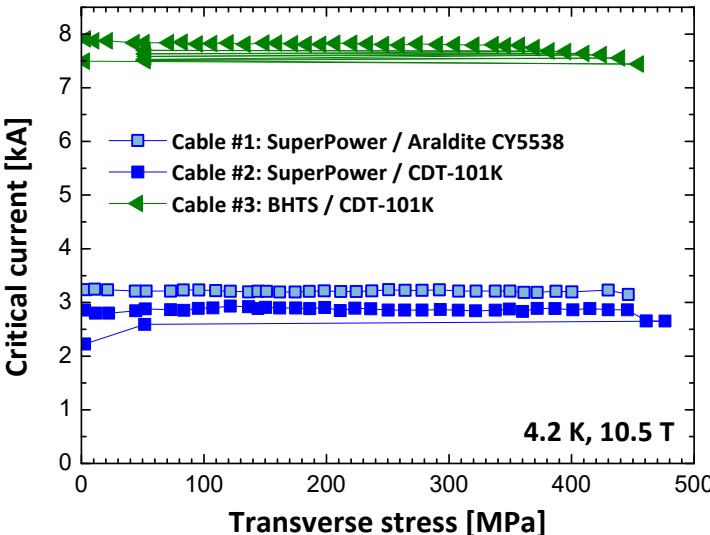

**Figure 15.** Critical current versus applied transverse pressure for three 15-tape REBCO Roebel cables: Cable #1 and Cable #2 are assembled from SuperPower tape and impregnated with Araldite CY5538 and CDT-101K, respectively. Cable #3 is made with BHTS tape and impregnated with CDT-101K. Adapted from Ref. [61].

### 5.3.3. AC Loss and Inter-Tape Resistance of REBCO Roebel Cables

EuCARD2 also promoted investigations on the magnetization, AC loss, and inter-tape resistance of REBCO Roebel cables [62–64]. Two measurement campaigns were carried out at the University of Twente and at the Southampton University on 15-tape SuperPower-type Roebel cable samples prepared at KIT and impregnated with CDT-101K at CERN. The experimental results were used as an input for an advanced electrical cable network simulation model proposed by Van Nugteren et al. [62], which was needed to analyze critical issues in the design of the magnets developed in EuCARD2, such as thermal stability, quench propagation, and dynamic magnetic field quality.

At Southampton University, AC loss inductive measurements were performed with an alternating magnetic field applied perpendicular to the broad face of the cable in a temperature range between 5 and 90 K and at a frequency of 5 Hz [63]. The main result was that the 15-tape Roebel sample behaves as two in-line magnetically coupled stacks, each stack of 7–8 tapes. Moreover, the AC loss is clearly hysteresis-dominated. This conclusion was confirmed by the group at the University of Twente: measurements were performed calorimetrically and inductively at 4.2 K and low frequency of 0.01–0.1 Hz, both in perpendicular and parallel field orientation. The coupling loss was found to be lower than the hysteresis loss in both orientations, within the range of the experiments [64]. However, these observations substantially differ from earlier results extracted from a similar cable but impregnated with the alumina-filled epoxy resin CTD-101G, which showed considerable coupling loss when exposed to a magnetic field parallel to the broad face of the cable [62].

The experiments from Gao et al. [64] were also complemented by direct measurements of the inter-tape resistance. A tape in a Roebel cable changes position along the longitudinal direction, thus within one transposition length every tape is in contact with two neighboring tapes. The inter-tape resistance is defined as the contact resistance over a single transposition length. It was found that its value at 4.2 K is in the range of 1–10 $\mu\Omega$, leading to a surface resistance of 0.5–10 $n\Omega m^2$, which is not significantly different from the values measured for the LTS wires in a Rutherford cable. These are encouraging results as they show that a balance is possible between coupling loss and current sharing to obtain thermal stability and adequate magnetic field quality.

## 6. Magnet(s) Design and Technology

The strong $J_c$ anisotropy vs. field direction, the large tape with the shielding of perpendicular field, and the intrinsic variation in the field inside the coil and the ramping regime require great effort for proper modeling. A quite detailed multi-physics e.m. model has been set up and is described elsewhere [65,66]. It determines the peak field for each direction and the critical surface in each point and the current distribution. This last is also computed during a quench, which is a major breakthrough in e.m. (electromagnetic) modelling, given the difficulty in protecting HTS magnets. This code has enabled the design and optimization of our EuCARD2 magnets.

### 6.1. Reference Magnet Design: AB Feather Magnets

The main objective of the magnet demonstrator was the validation of the HTS conductor by generating a 5 T dipole field in a free cold bore of 40 mm, as envisaged for high-energy LHC dipoles [9], the reference project at the start of EuCARD2. The design is based on a rectangular coil block dipole lay-out where the conductor (i.e., the wide face of the tapes composing the Roebel cable) in each block is aligned to the main field, the Aligned Block (AB) dipole FeatherM2 [65,66], see Figure 16. The field direction considered for the alignment is the one when the AB Feather magnet is excited inside the 13 T Fresca2 background field. The AB feather magnet generates a field, at a nominal current density of $J_e$ = 400 A/mm$^2$, of 5 T. Inside Fresca2 the total field should sum up to 18 T.

In stand-alone mode, this dipole should generate 5 T, too, despite the fact that the field lines are not perfectly parallel to the conductor wide face due to the different field

configurations. The extra margin due to lower field (5 T instead of 18 T) should compensate for the reduction due to non-negligible transverse field.

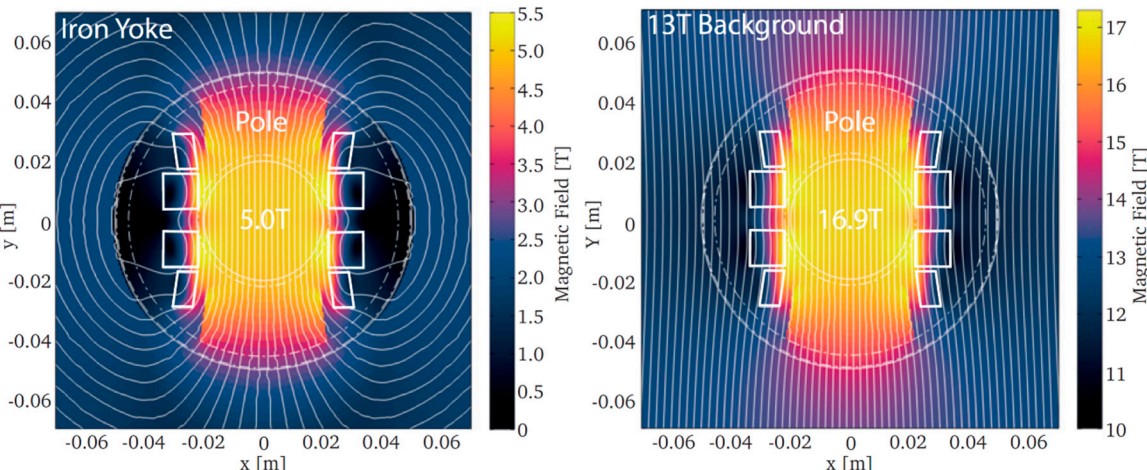

**Figure 16.** Cross-section of the coils of the Aligned Block dipole. The alignment is optimized when the dipole is inserted in the Fresca2 background field (see plot at the right), so is not as good as when the dipole is powered in standalone mode (left plot), [65].

The structure is based on an external support which is a pre-compressed stainless steel shell [59,67]; see Figure 17. A key feature is that the pre-compression does not need to completely counteract the e.m. forces during magnet excitation; because of the large temperature margin of HTS, small movements of the conductor should not lead to a quench. Therefore, the coil package can be inserted in the structure with some tolerance, making the assembly quite easy. The conductor during magnet excitation leans against the structure that has to be rigid enough to minimize the deformation. Interference with the Fresca2 structure is indeed forbidden by design. To make the structure rigid enough, avoiding excessive thickness of the outer restraining shell, the adopted solution was to link the outer shell to the inner structure with a midplane plate; see Figure 17. This reduces the free bore to 35 mm, but it is a temporary solution related to the test condition of Fresca2. There is no problem for force containment in a standalone test at 5 T, however at 18–20 T the forces are quite high. The e.m. model predicts that current is strongly non-uniform, because of the shielding properties of high-$J_c$ REBCO tape and—to a lesser extent—the partial coupling between tapes in the cable. A detailed analysis of the stress concentration due to high J and B, as well as to the non-uniform current distribution is reported in [68], where the mechanical structure of FeatherM2 is also described.

The coils are insulated with glass fiber braid that is sleeved on the cable before winding (the short length of the cable, less than 30 m, makes this operation quite easy). Then the coils are vacuum impregnated with epoxy resin (CTD 101K), carefully avoiding resin-rich zones to avoid delamination due to the high thermal contraction.

One further remarkable characteristic of the AB dipole is the use of copper rings in between coil layers and the outer shell. The ring imparts to the coil the radial force it receives from the restraining cylinder; however, most importantly, these rings are a well-coupled inductor to quickly extract the energy out of the coil following a quench and avoid an excessive hot spot temperature due to the low quench propagation velocity. The University of Tampere (Fin) has developed a model to evaluate the quench propagation for our HTS coils from basic properties [68,69], which greatly complemented the CERN e.m. numerical model.

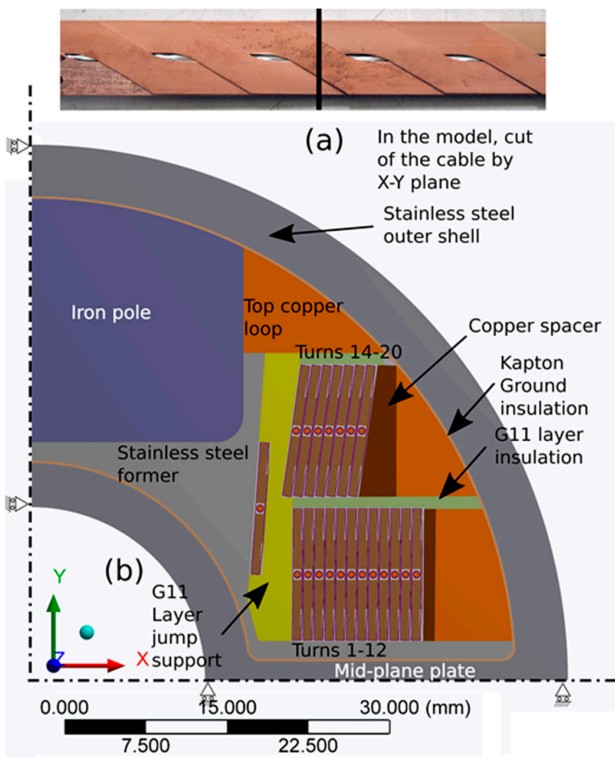

**Figure 17.** (**a**) Roeble cable used to wind the coils; black line indicates the cable section as it appears in the bottom picture. (**b**) One quarter cross-section of the AB FeatherM2 dipole (courtesy of J. van Nugteren and J. Murtomaki, CERN). The midplane steel connecting the inner and outer shell to control stresses at 18 T is indicated.

To gain experience with Roebel cable, the AB FeatherM2 dipoles were preceded by a smaller single coil winding, a six-turn simple flat racetrack, called FeatherM0 (with no cable alignment), see [65,66,70]. A sketch of FeatherM0 and Feather M2 is reported in Figure 18.

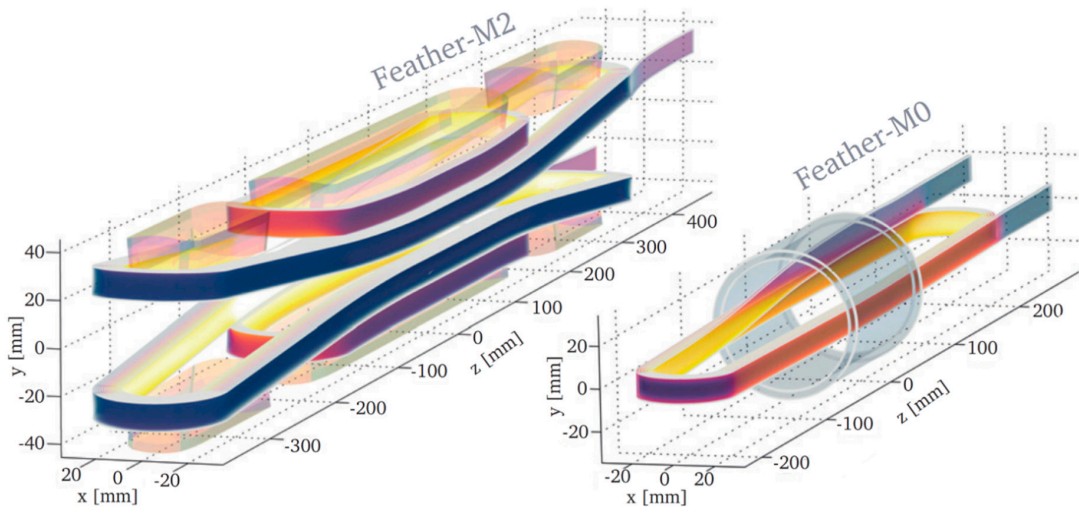

**Figure 18.** Sketch of the AB FeatherM2 dipole (**left**) and, in scale, of the coil test FeatherM0 (**right**).

Two FeatherM0 coils and two AB FeatherM2 dipoles were manufactured and tested; see Figure 19. The cable for winding two additional coils in order to assemble a third AB FeatherM2 has just now been completed.

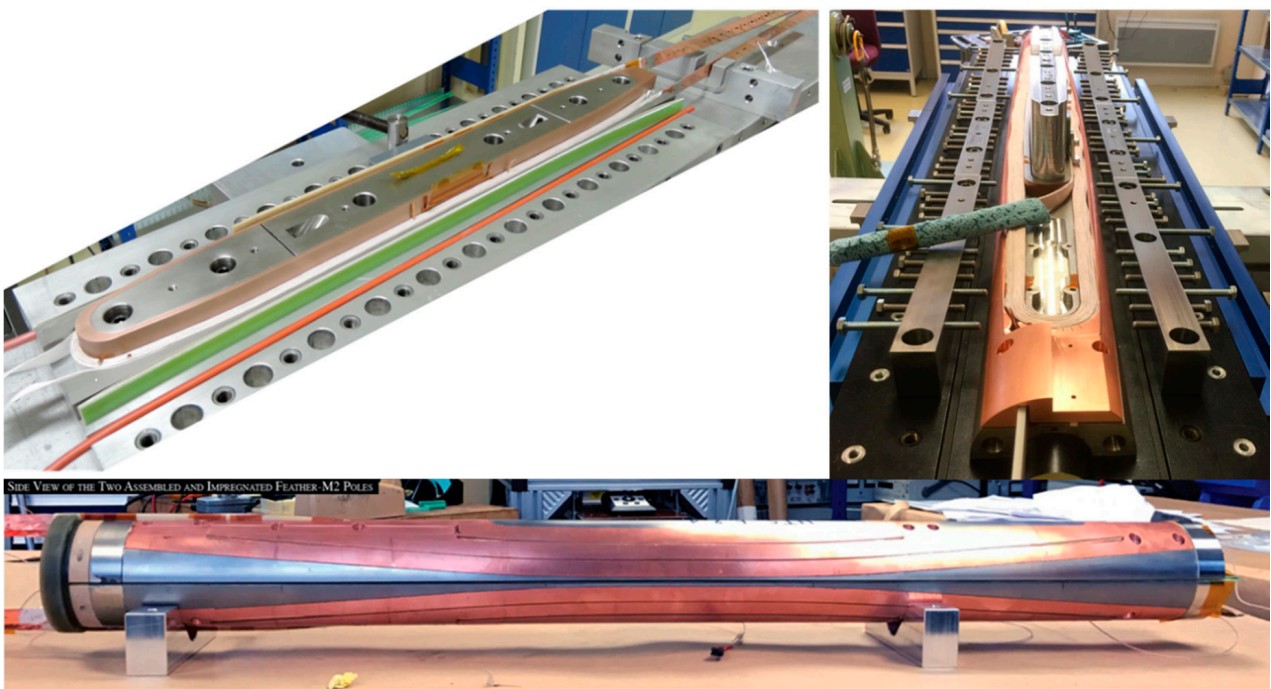

**Figure 19.** FeatherM0.5 coil after construction (**top left**). AB dipole FeatherM2.3-4 during winding (**top right**) and once assembled before yoking (**bottom**).

### 6.2. The EuCARD2 Cosθ Dipole

The EuCARD2 collaboration has also pursued the use of Roebel cable in the classical cosθ dipole configuration, the reference layout for accelerators [71]. The Cosθ configuration inherently has the largest field perpendicular to the wide face of the cable, a penalty for REBCO Roebel cable. However, this demonstrator is a good direct comparison between LTS and HTS technology. Since recent REBCO tapes have reduced the anisotropic behavior, thanks to various additional pinning mechanisms, though the "isotropization" is more marked at higher temperature, we think this is a very useful exercise. The effort is carried out by CEA, Saclay [72–74]; see Figure 20. The design is complex, both because of mechanical reasons and because Roebel does not behave as well as a Rutherford cable made of round strands. Additionally, the mechanical design for the high fields impeded the use of two layers since the forces were too high, so the coil is a single-layer cosθ. The coil ends turned out to be difficult and various tests and a full coil prototype with dummy cable were needed to work out this and other fabrication issues; see Figure 21. However, CEA has now overcome all issues, and the three coils (two needed and one spare) have just been wound and impregnated; see Figure 21. The insulation and impregnation technology is quite similar to that of the AB Feather dipoles. The Cosθ magnet assembly in its iron yoke is foreseen to end in autumn 2020, and a standalone test is scheduled at the beginning of 2021.

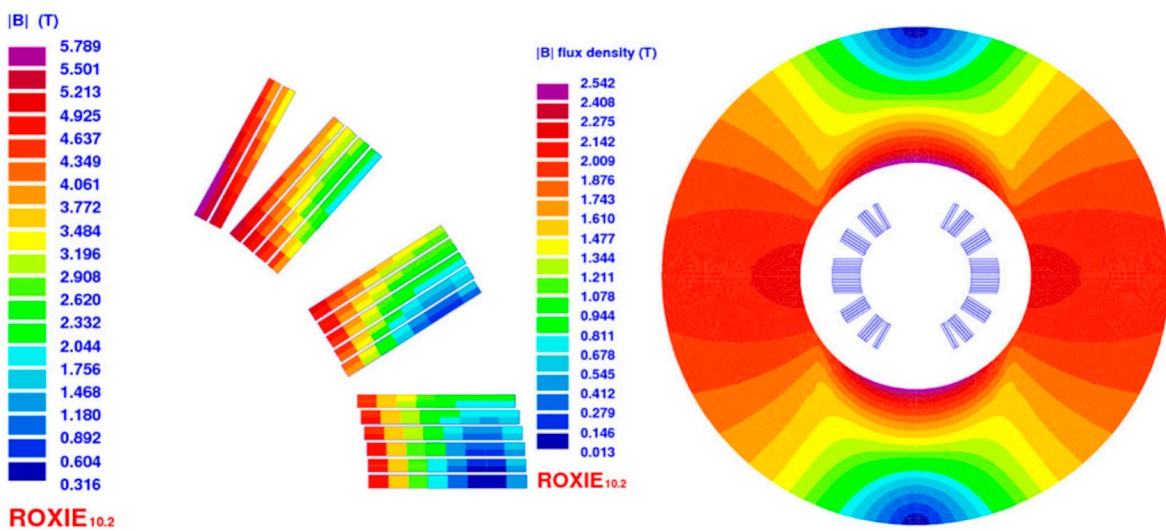

**Figure 20.** Cross-section showing the conductor layout (left) and the whole magnetic circuit of the CEA Cosθ dipole wound with Roebel cable for EuCARD2 (courtesy of M. Durante, CEA).

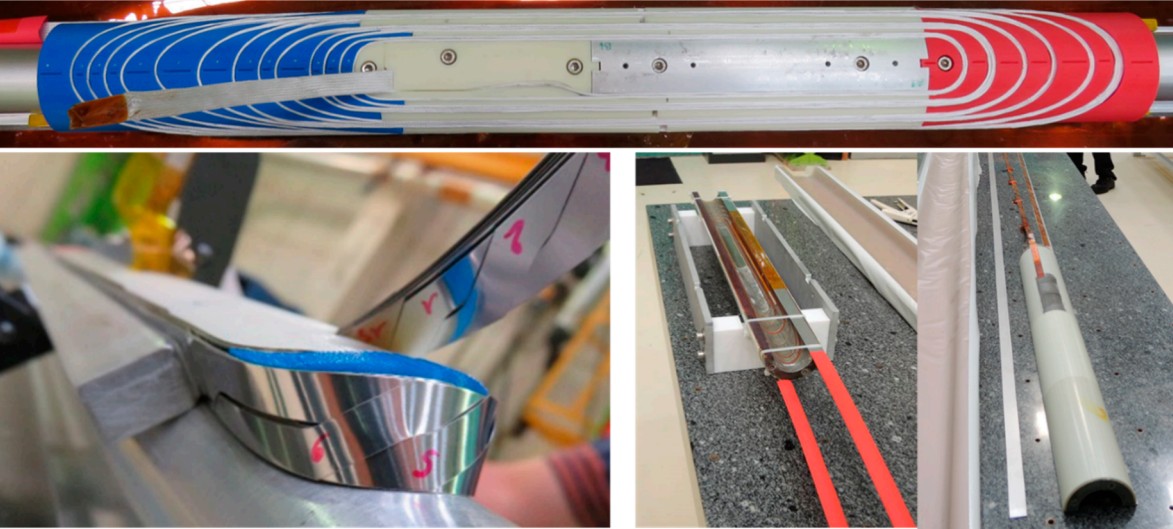

**Figure 21.** Top: practice winding of the CEA dipole with Roebel dummy cable (test with different end spacer concepts, the blue ones on the left have been finally chosen). Bottom left: detail of the cable turning during winding; center and right: pictures of the one of the three coils just after the impregnation (courtesy of M. Durante, CEA).

### 6.3. The EuCARD2 Stacked Tape Dipole

Another design was explored based on a cable composed of a simple stack of parallel tapes. This design has been pursued by INP Grenoble (Fr), based on a simple stack of 4 mm wide tapes, such as to have a square cross-section [75,76]. To avoid a difference in length among tapes, and to provide a minimum of transposition, the cable is twisted along its longitudinal axis right before and after the turning of the coil ends; see Figure 22. A full design has not yet been completed. In particular a more detailed analysis would be needed for mechanics, to limit the shear stress in the cable at the points where it is twisted, and for field quality. However, its simple structure, the high compaction factor, and the efficiency in use of HTS make this layout attractive as a possible alternative, once the basic issues are resolved.

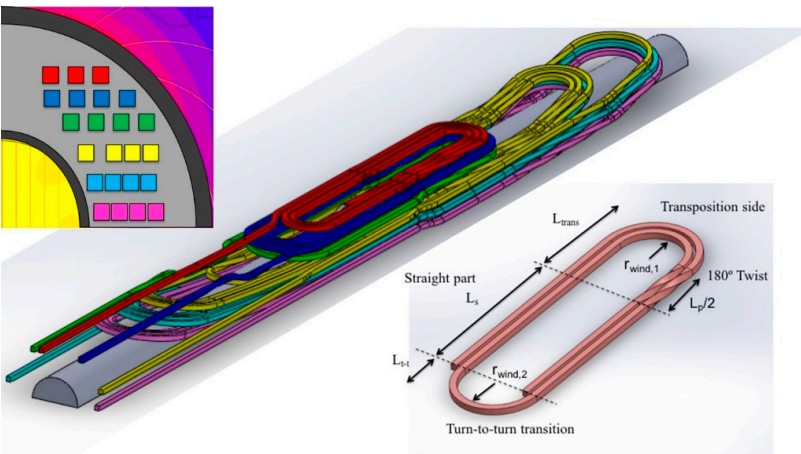

**Figure 22.** INP Grenoble dipole design with tape stack cable (4 mm tapes). Central: dipole coil layout (a racetrack with flared ends). Top left corner: one quarter dipole cross-section, each square is a tape stack—i.e., a 4 mm × 4 mm cable. Bottom left corner: cable twist of 180° in the winding to generate a tape transposition (courtesy of P. Tixador, INPG).

### 6.4. Magnet Technology

Three FeatherM0 coils were made from dummy cable composed of stainless steel tapes, punched like the HTS tape for Roebel, to learn about and test various technologies. An attempt to use glass ball-charged resin for impregnation failed, since the braid acts as a filter for the micro-balls (nano-size balls would be needed, which is quite complicated and expensive). We then used regular resin vacuum impregnation, taking care to minimize the resin-rich zone.

The first coil wound with superconducting cable, FeatherM0.4 [77], was used for benchmarking various diagnostic systems, especially for quench detection, which with current as high as 10 kA must happen in a few tens of ms. We developed a classical voltage comparison method, comparing the voltage inbalance between symmetric coils, with good sensitivity. The sensitivity was extremely high with the first dipole, FeatherM2.1-2, that had an insulated metal wire inside the cable, used as a reference loop perfectly coupled with the coils to cancel inductive signals, and good enough for the second dipole FeatherM2.3-4, which did not have such a coupled wire. Three additional types of sensors were placed in FeatherM0.4 to detect the quench: (a) a temperature array placed as near as possible to the coils; (b) an array of small flat pick-up coils capable of detecting the flux variation associated with current redistribution in tapes and cable; (c) a couple of optical glass fibers in a Bragg grating configuration to detect the temperature and/or strain status variation [78]. Optical fibers showed a good potential to detect a temperature rise well below the current sharing temperature. The other two systems were less successful, and a better position was identified for pick-up coils in the FeatherM2 magnet. In any case, classical voltage detection, with refined data acquisition, showed a sufficiently high sensitivity to detect the quench with sufficient anticipation to safely shut down the magnet even with large current (>10 kA).

An interesting feature of the FeatherM2 dipole design is the use, as mentioned above, of a copper ring around the coil as a structural element that acts as an inductive coupler capable of extracting a part of the energy in a very fast and safe mode from the coil. More details can be found in [79].

The first FeatherM0.4 test in 2016 [77] was extremely useful in improving connections with large transport current (10–15 kA) in conductors dominated by stainless steel, with little copper. Following the test on FeatherM0.4 a new type of connection was developed, called a "fin-block" joint, to capture each single tape with an appropriate transfer length, as discussed in [80]. However, the test was successfully concluded by injecting 13 kA in FeatherM0.4 at 12 K, after having first explored the critical surface from 80 K down

to low temperatures. The results were quite satisfactory, showing good current sharing among tapes.

## 7. Magnet Test and Results

All the tests so far have been carried out at CERN, where an adaptation of an existing set up allows testing from 80 K down to 5 K and then in liquid at 4.2 K, with current up to 20 kA. A standard and fast quench detection system (QDS) is available for these tests as well as a fast switch based on IGBT (insulated-gate bipolar transistor) technology to open the circuit in a few ms. From 2021, the INFN-LASA facility with similar characteristics to the one of CERN, will also be available.

Tests were carried out at CERN on the FeatherM0.4 coil at variable temperature that was then subsequently tested, also at various temperatures, in the high field split coil Sultan facility in the frame of a collaboration between CERN and EPFL-CRPP. A another test coil, FeatherM0.5, was tested directly in the Sultan facility. The two AB dipoles, FeatherM2.1-2 and FeatherM2.3-4, were tested in the CERN variable temperature facility.

### 7.1. Results on FeatherM0 Coils

The results of the first test at CERN on FeatherM0 are reported in Figure 23. The maximum current was limited by heating in the joints (from the coil terminals to current leads), rather than conductor in the coil. At each warmup, the configuration of the joints was modified and the cooling of the joints improved (by pouring LHe directly on the limiting joints). Finally, we could reduce the joint resistance down to the ten nΩ range and, at the fourth cooldown, we could see the coil quenching, see red solid square in Figure 23 In this way, we could validate the cable in terms of current transport since we exceeded the 10 kA limit. The cable itself was a very preliminary prototype out of the EuCARD2 collaboration. The coil was limited to 12 kA at 23 K, so we can extrapolate some 15–20 kA to be the quench limit at 4.2 K (our system was not suitable to sustain such high current in a safe mode). We did not observe degradation in the four cooldown–warmup cycles.

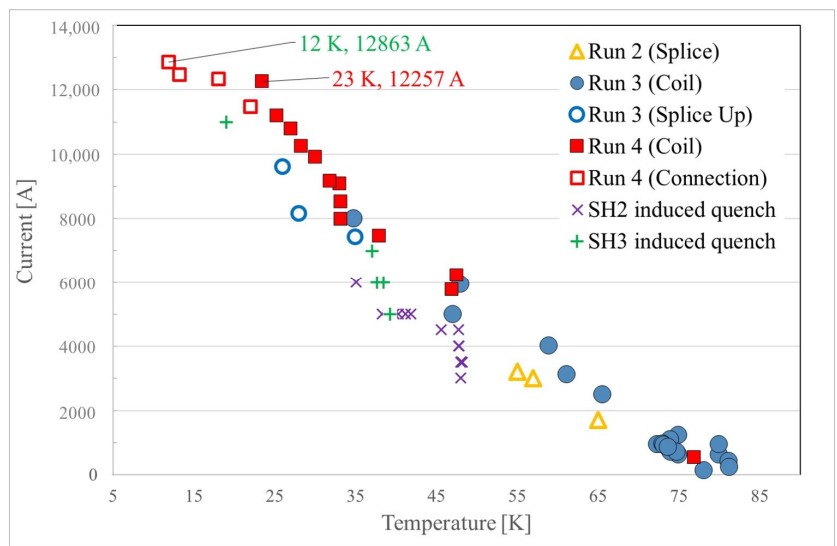

**Figure 23.** Results (maximum current vs. operating temperature) of the power test of FeatherM0 at CERN. The run numbering is accompanied by an indication of where the quench happened spontaneously (coil or splice). SH are heaters for the induced quench. Courtesy of J. van Nugteren and H. Bajas (CERN).

The coil was then tested in the Sultan facility of EPFL-SPC in Villigen (CH) [81]. After cooldown, the coil reached the same performance as at CERN, however with higher coil resistance (35 nΩ). Then, it reached 10 kA in steady state under an 11 T background field. Given the modest self-field of the coil (about 0.2 T/kA), the conductor was exposed to a

field of 13 T total. The internal resistance of the coils generated substantial heating. Then, in the subsequent warmup-cooldown cycle the resistance increased dramatically in such a way that it was no longer possible to energize the coil.

The same behavior, extremely high resistance, happened straight away at the first cooldown of the FeatherM0.5 coil, manufactured with tape not from EuCARD2. Therefore, this test coil never saw important transport current.

The reason for this behavior is not clear: the most likely hypothesis is that it may be a sign of degradation under field with delamination between the REBCO layer and the substrate. In the case of FeatheM0.5, it seems that delamination happened after cooldown. However, investigation is still underway at CERN to obtain a definitive answer.

### 7.2. Results of AB Dipoles: First Two FeatherM2 Magnets

Because of moving the production plant from one site to a new one, Bruker could not deliver the EuCARD2 tape on time for the first FeatherM2 dipole. Therefore, it was decided to wind the first EuCARD2 dipole with a cable quickly procured by CERN from SuperOx. The tape was manufactured by SUNAM and the Roebel cabling was performed by SuperOx with the final responsibility for the product [82].

The FeatherM0.4 test was important also to debug the test facility and, as mentioned above, to improve some critical issues, such as the HTS joints. This allowed a fast test of FeatherM2.1-2. The first test happened in April 2017 (see Figure 24), just in time to meet the deadline of EuCARD2, reaching a field of 3.35 T when powered by 6.5 kA at 5 K and above the expectations based on $I_c$ evaluation with scaling law (we do not have a direct $I_c$ cable measurement). Actually, the maximum steady-state field was 3.1 T. The extra 0.25 T is gained by allowing the coil to increase smoothly in temperature, up to a certain point when the total voltage increase starts to accelerate and the power supply needs to be shut off. This behavior points to a very soft transition, smearing the voltage increase, dominated by current-sharing mode. This is extremely useful because the quench starts smoothly and is easy to detect. Actually, since the heating and voltage increases are very reproducible, one may think to run the magnet in voltage mode rather than in current mode (to reach the maximum current).

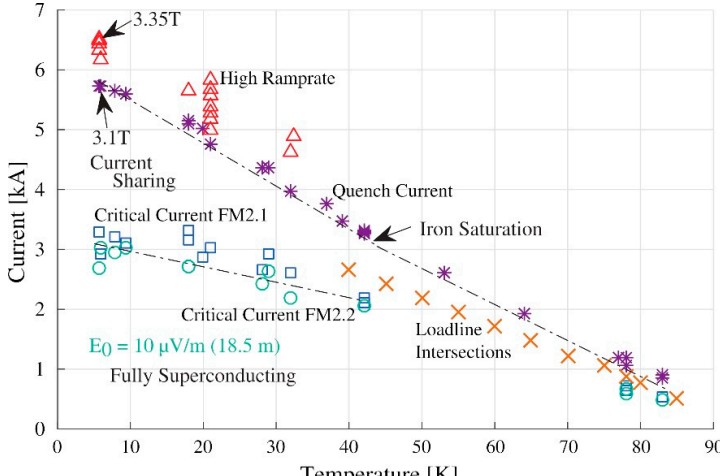

**Figure 24.** First test of the AB dipole FeatherM2.1-2 (quench current vs. operating temperature). Magnet excitation was carried out at a standard ramp rate of about 10 A/s, except when indicated as high ramp rate (hundreds A/s, red triangles). The region dominated by current sharing is indicated (between the two dot-dashed lines). Courtesy of J. van Nugteren and H. Bajas (CERN).

A full analysis of the test results is reported in [83,84]. The main points are:

- There is evidence of current sharing among tapes, as anticipated by the measurement of losses on cables. However, it seems that current sharing is larger than expected,

with contact resistance in the range of 10 μΩ. This helps to short-circuit local defects, but also makes the quench onset easy to detect.

- The transition is so smooth that we could increase performance by 10% and reach 3.35 T by increasing the ramp rate.
- The effect of energy dissipation in the copper ring, a feature that may be useful for future high-field magnets [79], is visible in accelerating the current decay.
- The absence of training—i.e., of repeated quenches to reach the maximum current for a given temperature [71]—is remarkable; even considering that it is a low field and small-energy magnet, this confirms the benefit of a large temperature margin.
- The first cooldown did not degrade the quenching performance. However, a subsequent thermal cycle saw a degradation of about 10% but the cooldown was not properly controlled. More verifications are under way.
- The first campaign of magnetic measurements showed a considerable number of field errors due to the coupling current among tapes, decaying with an 80 s time constant, and a relatively small contribution of intra-tape persistent current [85] of about 0.15–0.2% (15–20 units). The current decay might be triggered by low but not negligible internal resistance. While favorable from a field quality point of view, this last effect needs to be understood through more investigation.

Finally, the second EuCARD2 AB dipole, with the full performance tape of Bruker and the Roebel cabling by KIT, as from of EuCARD2 collaboration, was manufactured in 2018 and tested in 2019 and 2020. We performed three cooldown–warmup cycles. In the first two, we were limited by bad connections and probably by a damaged conductor in the leads just at the exit of the coils (probably due to bad manipulation). After some fixing, we were able, in the third test cycle, to reach 9 kA—i.e., 4.5 T in steady state—with a quench starting in the coil; see Figure 25. While very near to the design (5 T), this is visibly below the 6–7 T we could expect from the extrapolated short sample value on the tape. We do not have, as for FeatherM2.1-2, any direct measurement of $I_c$ in the cable. One good point is the absence of noticeable degradation after three cooldowns. The 9 kA limitation may actually not be due to the conductor in the coil; we suspect that the outer lead is heating (in the damaged point near the exit of the coil) and that the coil is driven into normal state by thermal conduction.

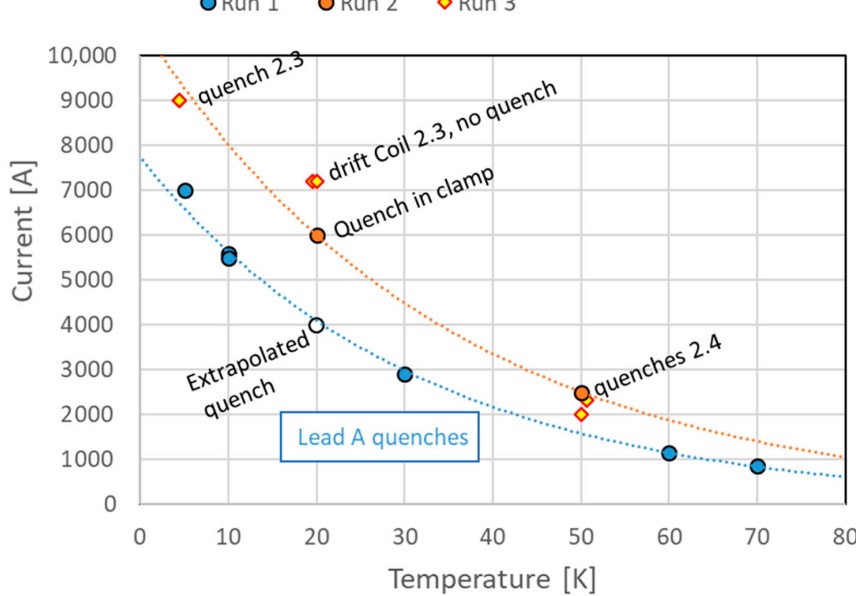

**Figure 25.** First powering campaign of the second AB dipole, FeatherM2.3-4, in terms of maximum current vs. operating temperature. Unpublished data CERN edms n. 2383644, by courtesy of G. de Rijk and G. Willering CERN.

Cold tests to assess the field quality will be resumed as soon as the COVID-19 pandemic is over. This HTS program is important for the long-term future of CERN, however it has, at present, very low priority with respect to LHC operation and high-luminosity LHC construction. Therefore, the reduced activity due to restarting after the COVID-19 lockdown means that this complementary program will be more severely delayed than other programs.

## 8. Conclusions

The European program EuCARD2, and to some extent also EuCARD and ARIES, have formed a small but continuous program on HTS accelerator magnets and conductor development. It has been instrumental in bringing together various laboratories and companies.

So far, we have demonstrated the capability of a high filling factor 10 kA class cable, Roebel type (up to a length of a few tens of meters), and developed a new dipole layout based on Aligned Block and flared ends to allow an aperture of 40 mm in the coils (reduced to 35 mm to allow insertion in the Fresca2 high-field facility without mechanical coupling). The highest field reached so far is 4.5 T in standalone with the FeatherM2.3-4. However, we hope with some modification to increase beyond this level.

In any case, we have two further magnets to test: a cosθ-dipole by CEA (not meant for record field but to learn about technology in a well-known configuration) and a third AB dipole, FeatherM2.5-6 by CERN, for which the cable has been procured according to the EuCARD2 specification. It should be manufactured and tested by spring 2021 and should go beyond the design goal of 5 T.

In general, we should say that HTS, despite the intrinsic difficulties in its use in accelerator magnets, keeps its main promise of training-free magnets and therefore offers solutions that should be exploited with novel designs and different approaches other than LTS.

Our program now also includes the testing of various inserts inside a high-field facility. The first one, FeatherM2.1-2 inside Fresca2, is underway. If one insert magnet can reach a field above the 16 T limit of LTS for dipoles, this would constitute a tremendous boost for HTS technology for particle accelerators and help in the design of even higher field dipoles [86].

**Author Contributions:** Conceptualization, L.R. and C.S.; methodology, L.R. and C.S.; validation, L.R. and C.S.; formal analysis, L.R. and C.S.; investigation, L.R. and C.S.; resources, L.R.; data curation, L.R. and C.S.; writing—original draft preparation, L.R. and C.S.; writing—review and editing, L.R. and C.S.; visualization, L.R. and C.S.; supervision, L.R. and C.S.; project administration, L.R.; funding acquisition, L.R. All authors have read and agreed to the published version of the manuscript.

**Funding:** This HTS program for accelerator magnets has been partly supported by EC mainly through the programs FP7-EuCARD2 Grant Agreement 312453 and H2020-ARIES Grant Agreement 730871. The main other sponsors providing contributions in hardware have been CEA, CERN, KIT, and the company Bruker HTS.

**Institutional Review Board Statement:** Not applicable.

**Informed Consent Statement:** Not applicable.

**Data Availability Statement:** Data sharing not applicable.

**Acknowledgments:** The authors acknowledge all EuCARD2-WP10 and ARIES task 14.5 collaborators. In particular, thanks are due to A. Ballarino, H. Bajas, L. Bottura, G. Kirby, F. Mangiarotti, J. Murtomaki, J. van Nugteren, G. de Rijk (CERN); M. Durante, C. Lorin (CEA); W. Goldacker and A. Kario (KIT); U- Betz and A. Usoskin (BHTS) for their major contributions. L. Rossi, having left CERN on the 1 October 2020, wants to express his gratitude to all the CERN team that has collaborated in the HTS program in the frame of the EU programs in the years 2014–2019.

**Conflicts of Interest:** The authors declare no conflict of interest.

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
