# Peer review of "HTS Accelerator Magnet and Conductor Development in Europe"

_instruments, doi:10.3390/instruments5010008_

Round 1

Reviewer 1 Report

This paper presents a nice review of the efforts to develop HTC magnets fit for acceleration purpose over the past 15 years. It is a summary liable to interest an enlarged accelerator community, or even other physicists interested to get a synthetic information on recent progress.

After correction of a few typo/language errors, it could be published as such. Nevertheless, I have some further suggestions addressing the legibility for a larger public:

P7; l.283-287: an additional figure showing the various type of cable would help the non-specialized readers.

P15; l.501: fig 8 is obviously fig 12

P16; fig.13: In this context, “punching” is not a familiar term for non-specialized readers. Completing the figure with the scheme of a single tape and its punching delimitation would make it more self-understandable.

P19; l.607: fig 10 is obviously fig 16

Please check other numberings

P19; l.668-669: oral turn of phrase, change “easy job”

P22; l.727: “bene”=> “been”

P25; l.800: precise where Sultan facility is.

P26; l.843: “training” is not a familiar term for non-specialized reader, it would be better to define it somewhere (e.g. foot note)

P26; l.848-850: I do not understand the meaning of this sentence.

P26; l.858: Oral turn of phrase. “After some fix was put in place, we were able…”=> “After some fixing, we were able…”.

P27; l.884: Oral turn of phrase. “But we hope with some fix to increase”=> “some improvement” or “some modification”… At least “some fixing

Further suggestion: the English level of the first 2 pages is different from the rest of the document. Have some English native speaking person review it.

Author Response

All suggestions and requests have been  addressed and answered.

Reviewer 2 Report

The paper ‘HTS Accelerator Magnets and conductor development in Europe’ is a complete an easy to read review of the activities of the EuCARD2 and ARIES programs and beyond. It’s of high interest for the community developing superconducting magnets for the accelerator of next generation, and shows the very important role and leading played by CERN with respect to other institutes and companies in Europe.

My recommendation is to publish it mostly in the present form adding few corrections of typos and explanations. I am sure that I did not catch them all the typos. A final check should be made by the authors and editor. I also recommend top add few explanations to help a reader which is not working on the development of superconducting dipoles for accelerators. Please find my recommendations for changes below:

Line 20: a bracket is missing.

Line 23: please explain what is meant with race track with no bore exceeded 5 T. Where is the 5 T reached?

Line 35: ‘8.3 Tesla’ -> ‘8.3 T’

Line 56: ‘tunrs’ ->’turns’

Lines 61-63: Please explain how the dipoles increase the beam intensity and their relation to the collimators.

Line 108: As in the abstract a racetrack coil with no bore is mentioned. Please explain what is meant with this. Is it meant a coil structure as shown in Fig. 4? How does this relate to the final coils with bore?

Line 114: shouldn’t 70 µm be 60 µm, as shown in Fig. 3?

Figure 4: please show where the 5.4 T have been reached.

Line 143: Please explain how the magnetic field is measured.

Line160: ‘base’ -> ’phase’

Line 202: ’temperature.)’ ->’temperature)’

Line 210: ‘taper’ -> ‘tape’

Line 218: Please make an example to explain about which ‘accelerator characteristics’ you mean.

Lines 232-233: Please add a sketch to show the 3 types of magnets or refer to the section where they are described.

Line 240: Please spell out the acronym INPG.

Line 264: Please add more detail on the buffer layer: what is it made of?

Table 1: please define the different parameters (σ, µ0, etc…)

Line 319: Please add some info on the required bending radius of the HTS tape.

Line 403: Please state the length of the tapes.

Figure 8, caption: please explain what is meant with 2x20µm Cu stabilizer. Is it on both sides? Please add a description in the text as well.

Figure 9: please define the parameters of all axis.

Line 448: ‘manufacturer another’ -> ‘manufacturer to another’

Figure 10: please define all the parameters described in the label.

Figure 11: please clarify if the stress is always longitudinal tensile or compressive as well.

Line 511: Please remind the reader how the RRR depend on the magnetic field.

Lines 746-748: please indicate if a solution, and in cased yes which, has been found for the impregnation.

Line 751: Please explain between which voltages the comparison is made.

Line 769: Please show a picture or sketch to describe the Fin-Joint or use the correct reference.

Line 777: Please spell out the acronym IGBT.

Line 788: Please explain what is meant in the legend of Figure 22.

Line 801: ’(35 nΩ.’ -> ’(35 nΩ).’

Line 827: This statement is not clear to me. Normally the quench detection in HTS magnets is considered to be more difficult because of the smaller velocity of the quench propagation with respect to LTS magnets. This statement seems to contradict this. Please explain.

Figure 24: please clarify the labels.

Author Response

All questiosn and suggestiosn have been addressed.

Round 2

Reviewer 2 Report

I reccommend to publish the paper in the present form.